# Instantly adhesive and ultra-elastic patches for dynamic organ and wound repair

Parth Chansoria[1,9], Ameya Chaudhari[1,9], Emma L. Etter[1], Emily E. Bonacquisti [1], Mairead K. Heavey[1], Jiayan Le[1], Murali Kannan Maruthamuthu[1], Caden C. Kussatz[1], John Blackwell[2], Natalie E. Jasiewicz[1], Rani S. Sellers[3], Robert Maile[4,5], Shannon M. Wallet[6], Thomas M. Egan[2,7,8] & Juliane Nguyen [1,7] ✉

Bioadhesive materials and patches are promising alternatives to surgical sutures and staples. However, many existing bioadhesives do not meet the functional requirements of current surgical procedures and interventions. Here, we present a translational patch material that exhibits instant adhesion to tissues (2.5-fold stronger than Tisseel, an FDA-approved fibrin glue), ultra-stretchability (stretching to >300% its original length without losing elasticity), compatibility with rapid photo-projection (<2 min fabrication time/patch), and ability to deliver therapeutics. Using our established procedures for the in silico design and optimization of anisotropic-auxetic patches, we created next-generation patches for instant attachment to tissues while conforming to a broad range of organ mechanics ex vivo and in vivo. Patches coated with extracellular vesicles derived from mesenchymal stem cells demonstrate robust wound healing capability in vivo without inducing a foreign body response and without the need for patch removal that can cause pain and bleeding. We further demonstrate a single material-based, void-filling auxetic patch designed for the treatment of lung puncture wounds.

Bioadhesive materials and patches have attracted considerable academic and commercial interest as replacement materials for sutures, staples, and wound dressings[1–3]. While sutures and staples remain the current standard approach for closing wounds and cuts, they do not always prevent leakage and can contribute to surgical wound complications, toxicity[4], inflammatory responses[5], undesirable matrix remodeling and scar tissue formation[5], and post-operative pain[6]. While staples are a valid alternative to sutures and can be used in mass casualty situations due to their quick application, they are not applicable in every situation. For instance, they are usually not suitable

for closing wounds on the hands and feet because they can cause discomfort[7].

These limitations of current approaches have prompted interest in using bioadhesive materials and patches for wound closure, wound healing, air leak management, and tissue sealing. However, many existing bioadhesives do not meet the functional requirements of modern surgical procedures and interventions. For example, liquids and glues are easily displaced and diluted when applied to wet tissues and/or internal organs. Other bioadhesives become ineffective and completely lose their tissue adhesiveness in wet environments or when

[1]Division of Pharmacoengineering and Molecular Pharmaceutics, Eshelman School of Pharmacy, University of North Carolina at Chapel Hill, Chapel Hill, NC 27599, USA. [2]Division of Cardiothoracic Surgery, Department of Surgery, School of Medicine, University of North Carolina at Chapel Hill, Chapel Hill, NC 27599, USA. [3]Pathology and Laboratory Medicine, Department of Medicine, University of North Carolina, Chapel Hill, NC 27599, USA. [4]Department of Surgery, University of North Carolina at Chapel Hill, Chapel Hill, NC 27599, USA. [5]Department of Microbiology and Immunology, School of Medicine, University of North Carolina at Chapel Hill, Chapel Hill, NC 27599, USA. [6]Division of Oral and Craniofacial Health Sciences, University of North Carolina at Chapel Hill, Chapel Hill, NC 27599, USA. [7]Joint Department of Biomedical Engineering, University of North Carolina at Chapel Hill, Chapel Hill, NC 27599, USA. [8]North Carolina State University, Raleigh, NC 27695, USA. [9]These authors contributed equally: Parth Chansoria, Ameya Chaudhari. ✉e-mail: julianen@email.unc.edu

they encounter bodily fluids. While earlier bioadhesive materials required at least several minutes of tight tissue-patch interfacing for effective tissue adhesion[8,9], recent advances have demonstrated quicker and more rapid bioadhesion to dry and wet tissues[10,11].

However, barriers to clinical translation for bioadhesive materials still exist and include insufficient flexibility and an inability to conform to the mechanics of internal organs and the skin. Dynamic organs including the lungs and heart present specific challenges towards the effective use of tissue adhesives and patch-based therapeutics[1]. Their repetitive physiological motion and volumetric expansion and contraction create a fatigue loading environment that predisposes patches or bioadhesives to premature breakage or detachment[1,12]. Patches are typically held in place using sutures, staples, or surgical glues that can similarly detach or strain the organ over time, require specialist skills to apply, and present post-operative complications (inflammation, scarring, etc[5].). For external skin applications, it is crucial that the bioadhesives or patches provide sufficient flexibility and elasticity to withstand tension caused by twisting and bending and remain attached. Furthermore, to avoid discomfort and pain from repeated dressing changes, patches should also be made to biodegrade and disintegrate during the period of administration[13,14].

In many applications, applying a bioadhesive patch alone does not promote tissue healing, so delivering therapeutics and drugs via bioadhesive materials and patches would be desirable to accelerate the healing process. Here, recent advancements with functionalized hydrogels have shown promising potential for controlled drug delivery in vivo[15,16]. Accordingly, we posit that an ideal bioadhesive or patch should be: (1) biocompatible and biodegradable, (2) strongly and rapidly adherent, (3) elastic and mechanically conform to the underlying tissue, and (4) able to deliver therapeutics to accelerate healing.

In our previous work, we established a design framework for developing hydrogel patches with anisotropic and auxetic properties[17]. However, this initial set of materials had limited stretchability and were not intrinsically tissue adhesive. Attaching the patches to tissues required the application of fibrin glue, which is expensive so precludes their use in low-resource settings. Furthermore, the use of fibrin glue for adhesion could introduce variability during the preparation and application of the glue, potentially affecting the mechanics of the patches. For consistent clinical implementation, prefabricated adhesive patches are more suitable, as they can be prepared in a controlled environment and provided to the clinics for immediate use.

Therefore, the objective of this work was to develop the next generation of patch materials with the following characteristics: (i) instantaneous and strong tissue adhesion; (ii) ultra-elasticity, (iii) adaptable and scalable printing into precise anisotropic-auxetic architectures for a wide range of organs and tissues, (iv) biocompatibility and biodegradability, and (v) ability to deliver therapeutics. We applied our previously established framework for creating anisotropic-auxetic architectures[17] to develop an ultra-elastic and instantly adhesive patch platform adaptable to a broad range of tissue-specific mechanical properties, including elasticity, stiffness, and rigidity. Here, we demonstrate how our patch system can be readily fine-tuned for instantaneous attachment to different organs and conform to organ mechanics ex vivo and in vivo. Our patches show substantially stronger attachment to wet tissues compared with commercially available fibrin-based Tisseel® (Baxter). Lastly, we evaluate the therapeutic potential of the patches in two exemplar pathologies, wound healing and pulmonary air leakage. Here, the adhesiveness of the patches allows coating with extracellular vesicles (EVs) for enhanced wound healing without the need for patch removal and without inducing a foreign body response. With respect to pulmonary air leakage, we demonstrate how a single patch material can be fine-tuned to prevent air leakage from a punctured lung while conforming to the lung's auxetic mechanics.

## Results

### AuxES patches - material composition and patch fabrication

Our platform of **aux**etic, **e**lastic, and **s**ticky (AuxES) patches, is based on a composition of materials that when combined lead to its high elasticity, adhesiveness, and high-resolution lattice structure. Notably, the materials on their own, including gelatin methacryloyl (GelMA), acrylic acid (ACA), calcium chloride (CaCl₂), and curcumin nanoparticles (CNPs) do not deliver the desired characteristics, but only when combined are able to conform to dynamic organ mechanics and achieve therapeutic functionality. The patch fabrication procedure shown in Fig. 1A is simple, cost-effective, rapid, and highly reproducible. The material can easily be formulated using off-the-shelf reagents (all reagents can be commercially procured) and is selectively photocrosslinkable, rendering it compatible with digital light projection (DLP), which has specific advantages over other conventional or advanced manufacturing techniques[18]. In DLP, a digital micromirror device is used to project the image of a construct one layer at a time, which allows facile fabrication of highly complex architectures, which is otherwise challenging using conventional techniques such as casting or electrospinning[18].

We chose gelatin as the polymer backbone for the matrix as it is widely studied and has good biocompatibility, possesses integrin binding sites (RGD) for cell adhesion, and can undergo enzyme-mediated biodegradation[19–21]. GelMA is prepared through the methacrylation of gelatin and can undergo rapid chain growth polymerization in the presence of a photoinitiator, providing excellent compatibility with high-resolution photocrosslinking[22,23]. GelMA has become the gold-standard for bioprinting in recent years due to its easy and replicable production, however on its own does not provide tissue adhesion[23,24]. We used LAP (lithium-phenyl-2,4,6-trimethylbenzoylphosphinate) photoinitiator (0.03% w/v) due to its compatibility with visible UV light (405 nm) crosslinking and biocompatibility[20,25]. The addition of ACA did not affect the photocrosslinking of the matrix formulation, and the use of a collimated light beam in DLP allowed patch fabrication as a single layer in under 2 minutes[24], substantially faster than extrusion printing[18].

CNPs act as a UV-absorptive and radical quenching agent critical for high-resolution DLP fabrication[24]. We optimized the matrix CNP concentration by determining the print resolution and thickness of template patches with a broad range of feature sizes (Fig. 1B) made using different CNP concentrations. Increasing the CNP concentration significantly reduced the minimum achievable feature size ($p < 0.05$) but also reduced patch thickness by affecting the UV penetration depth. We finally selected 2 mg/ml CNP, since it achieved a minimum feature size of 375 μm and a consistent patch thickness of ~1.2 mm, which produced stable patches (Fig. 1B) Before commencing experiments, the patches were dipped into a solution containing 0.5 M CaCl₂ and 0.05 M NaOH in deionized (DI) water. For wet tissue application, i.e. the application to the lungs, we observed that air drying improves the adhesiveness of the patches. This may be due to increased water absorption capacity and hydrogen bond formations between the tissue and the patch, however no air drying was necessary for application on dry tissue such as the skin (see Supplementary Fig. 1A for the effect on drying time).

For any implantable medical device, the demonstration of lack of cytotoxicity is an essential first step. We therefore evaluated the viability of 3T3 fibroblasts cultured for over 48 h in the presence of the patches. In addition to CNP-laden patches, we also tested CNP-free patches made using FD&C yellow food dye, which has previously been shown to demonstrate the radical quenching needed for high-resolution DLP[24]. The presence or absence of CNP had no effect on cell viability (~100% after 48 h, Fig. 1C).

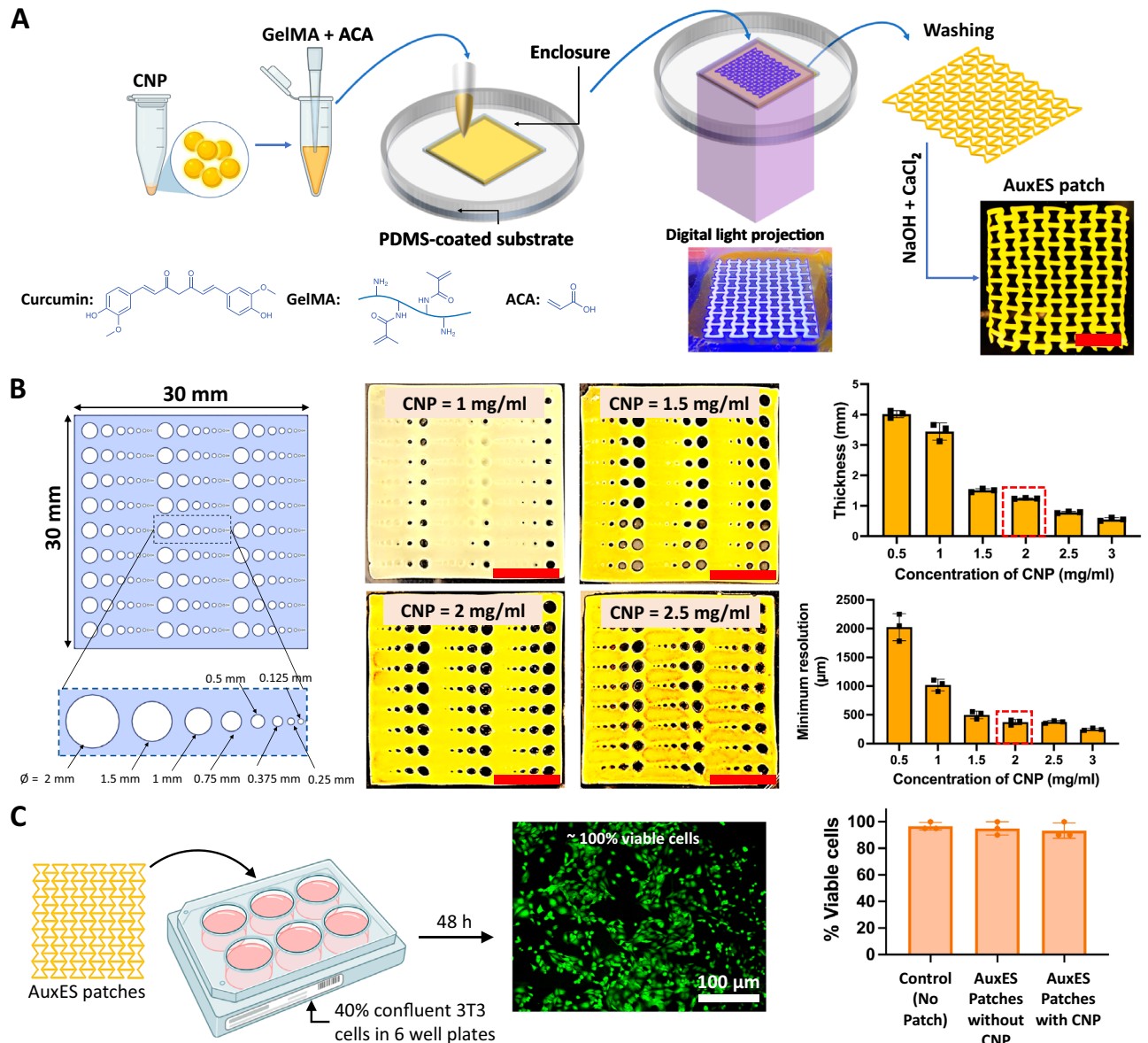

**Fig. 1 | Auxetic elastic and sticky (AuxES) patches - fabrication and printing optimization. A** The AuxES patches are fabricated via a facile single-layer projection-printing of a photoink (GelMA + ACA) contained in a PDMS-coated substrate. After washing and neutralization, calcium chloride is used to enhance patch integrity and adhesion. Panel A was in part created with Biorender.com. **B** Optimization of CNP content within AuxES patches for high-resolution printing. Since CNPs act as a radical quenching agent, increasing their concentration improves print resolution. However, it also reduces the UV penetration depth within patches (for the same UV exposure duration). A 2 mg/ml CNP concentration was chosen since it results in high print resolution (~375 μm minimum feature size) and consistent thickness of ~1.2 mm. **C** The patch materials demonstrate high biocompatibility with or without CNP as tested with viability of 3T3 cells after 48 h in culture. Panel C was in part created with Biorender.com. Data in **B** and **C** are from $n = 3$ biologically independent samples and are presented as mean ± s.d.

## Patches demonstrate instantaneous adhesion to wet tissues

A material with strong and instantaneous bioadhesion can substantially reduce the time needed for patch application, thereby improving acceptance from surgeons and minimizing intraoperative harm[1]. Attachment tests on ex vivo organs (Fig. 2A demonstrates attachment over a mouse liver; excess fluid over the tissue was removed through gentle dabbing using a blotting paper) using our AuxES patch formulation demonstrated strong and instantaneous (<1 s) bioadhesion to wet organs and could support the entire weight of a mouse liver upon immediate contact (see Supplementary Movie 1). We further measured the forces required to detach the patches attached to mouse livers (Fig. 2B), and the adhesion force for AuxES patches was almost twice that ($p < 0.05$) of commercially available

fibrin glue (TISSEEL, Baxter Healthcare) and ~10-times that of GelMA patches. When stretched, AuxES patches broke but did not detach from the organ (Supplementary Movie 2), further demonstrating the strength of bioadhesion. Of note, different patch architectures may exhibit different adhesiveness, which is related to patch area in contact with the organ and the viscoelastic properties (Supplementary Fig. 1C–E). Importantly, the AuxES patches not only strongly adhered to tissues but also imparted the ultra-elasticity needed to conform to various dynamic organs (Supplementary Movie 3). The strong bioadhesion of the patches can be attributed to a combination of hydrogen and ionic bonding between the patch and the underlying tissue matrix (Fig. 2C). Addition of acrylic acid to GelMA increases the number of carboxylic acid groups in the photocrosslinked matrix[26] and leads to

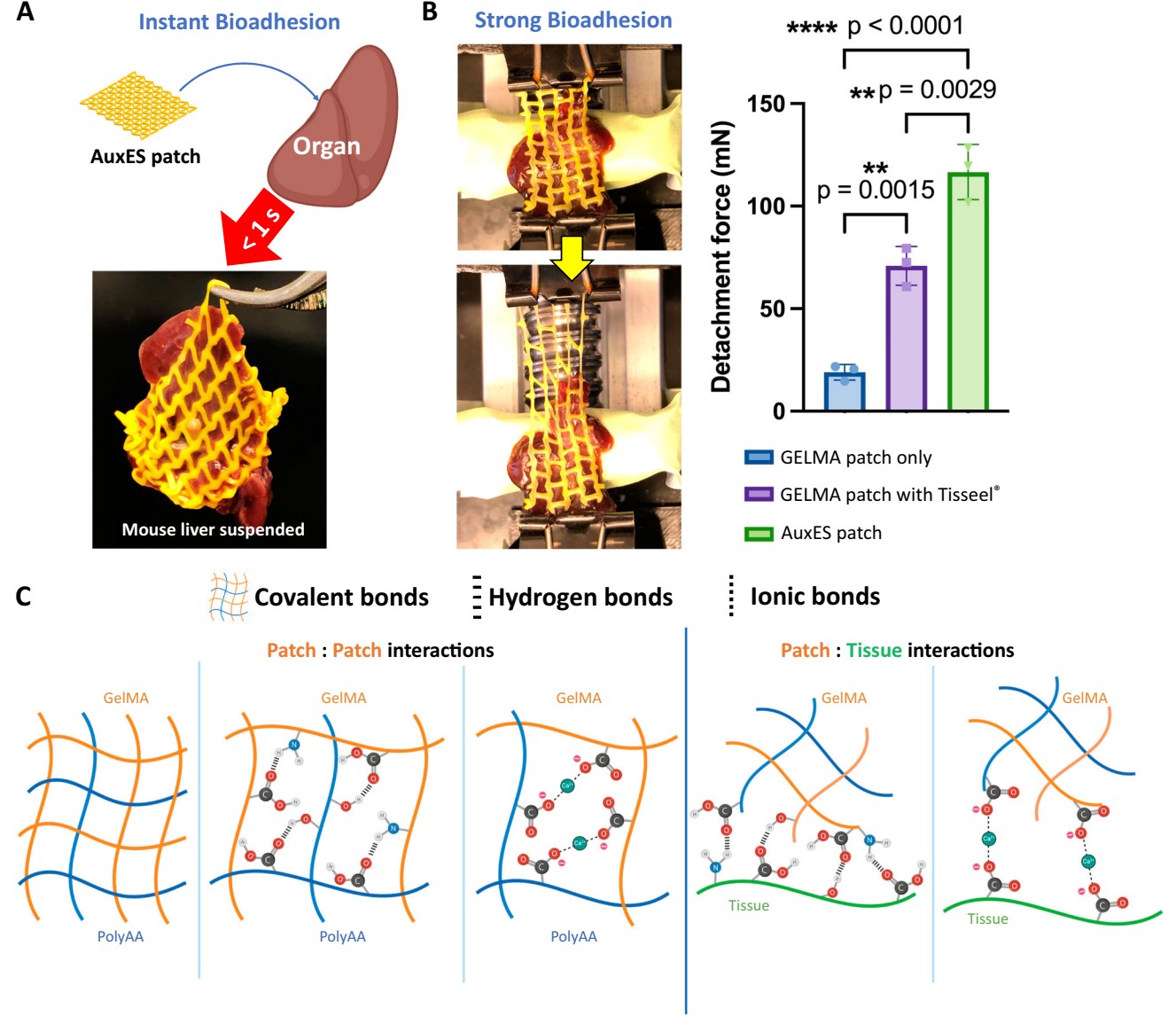

**Fig. 2 | *Bioadhesive properties of the AuxES patches. A.*** Patches attached to a healthy mouse liver demonstrate instantaneous adhesion in <1 s sufficient to support the weight of the mouse liver (also see Supplementary Movie 1). Panel **A** was in part created with Biorender.com. **B** Patches demonstrate strong bioadhesion to the mouse liver. The detachment forces for the AuxES patches (also see Supplementary Movie 2) are 10-fold higher than GelMA alone and robustly higher than those for a GelMA patch attached with an FDA-approved fibrin glue (TISSEEL, Baxter). Data are from $n = 3$ biologically independent samples and are presented as mean ± s.d. Statistical significance was assessed by one-way ANOVA followed by Tukey's multiple comparison tests, *$p < 0.05$, ** $p < 0.01$, ***$p < 0.001$, and ****$p < 0.0001$. **C** Schematic showing that the combination of hydrogen, covalent, and ionic bonding imparts instantaneous and strong bioadhesion and ultra-elasticity to AuxES patches. PolyAA = polyacrylic acid. Panel **C** was in part created with Biorender.com.

the formation of polyacrylic acid (polyAA), which enhances patch bioadhesion to wet tissue through absorption of interfacial water and hydrogen bond formation, while calcium chloride results in ionic crosslinking of patches with the carboxylic acid groups in tissue matrix to further improve bioadhesion (see Supplementary Fig. 1B for adhesion force measurements with different ACA content). The importance of $CaCl_2$ treatment post patch printing is also highlighted in Supplementary Movie 2, where patches without $Ca^{2+}$ crosslinking are more easily detached from the organ, while patches with $Ca^{2+}$ crosslinking do not detach from the organ. The prevalence of ionic and hydrogen bonding, in addition to the covalent binding of methacrylate moieties (Fig. 2C), also improves the stretchability of the matrix (Fig. 2A, Supplementary Movie 3), which is discussed in further detail in the subsequent section.

## AuxES patches are designed to adapt to the mechanics of dynamic organs

Dynamic organs have highly variable anisotropy and auxetic or non-auxetic characteristics. For example, the lungs, myocardium, bladder, stomach, intestines, diaphragm, and the tendons are dynamic organs, of which the diaphragm and tendons are non-auxetic (positive Poisson's ratio of surface deformation), while other organs are auxetic (negative Poisson's ratio of surface deformation)[1]. For the design of the AuxES patches, we considered intrinsic organ anisotropy, where the organ stiffness (E) is different in the longitudinal (assumed as the predominant direction of ECM fiber orientation) and transverse (direction perpendicular to the ECM fibers) directions. Since the void space between the ECM fibers reduces the resistance to deformation in the transverse direction[1], the longitudinal:transverse stiffness ratio

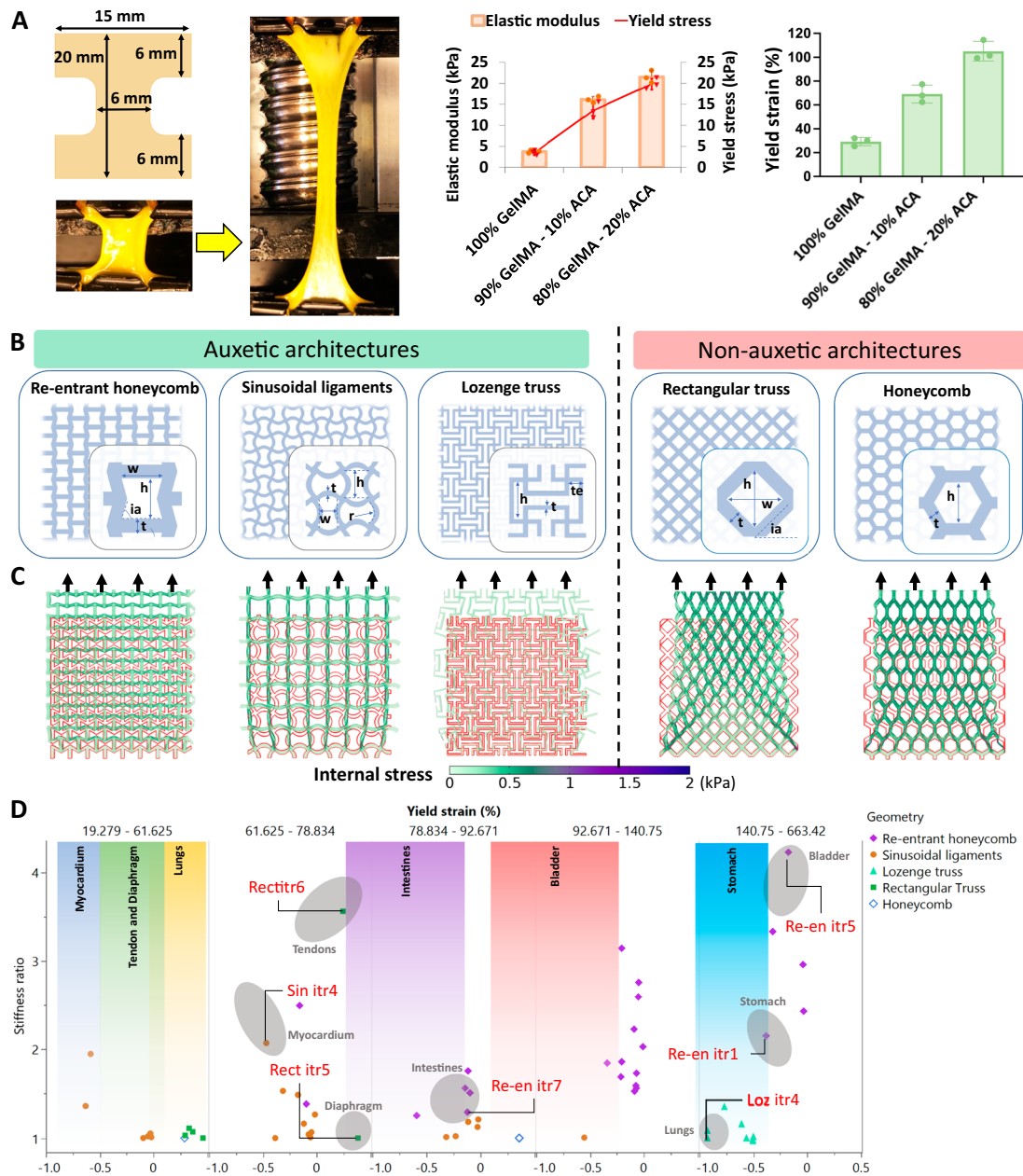

**Fig. 3 | Analytical and computational modeling to optimize AuxES patch design for different organs. A** Different dog bone-shaped patch formulations first underwent tensile testing to optimize the patch formulation and its mechanical properties for computational modeling. 80% v/v GelMA − 20% v/v ACA demonstrated the highest stretchability (~110% yield strain) and elastic modulus (~ 20 kPa) and was used in the computational models. Data are from $n = 3$ biologically independent samples and are presented as mean ± s.d. **B** Selected auxetic (re-entrant honeycomb[12], sinusoidal ligament[31], and lozenge truss[32]) and non-auxetic (rectangular truss[32] and honeycomb[33]) architectures were used within the computational models. Design features (w, h, r, ia, t, te) were varied (see Supplementary Table 1 for the values) to generate a library of over 60 patch designs. **C** Computational outputs depicting internal stresses within the selected iterations of the geometries of auxetic (negative Poisson's ratio) and non-auxetic (positive Poisson's ratio) architectures using the 80% GelMA − 20% ACA formulation. Note that the auxetic patches expand laterally when stretched, while non-auxetic patches contract laterally when stretched. **D** Plots of the patch stiffness ratios, Poisson's ratios, and yield strains and overlap with the properties of individual organs. The yield strain plot lists the maximum linear strain that the organ undergoes in its normal physiological cycle. The patch fit to each dynamic organ was decided based on both the stiffness curves and Poisson's ratios. The selected patch architectures are further verified such that yield strain of the patch is always greater than that of the normal physiological strain of the organ.

(henceforth referred to as $E_L$:$E_T$) is greater than 1. Except for the lung, which is mostly isotropic ($E_L/E_T \sim 1$), other organs feature anisotropy ($E_L/E_T > 1$)[1].

To determine the mechanical characteristics of AuxES patches, structural mechanics computational models were developed to simulate unidirectional (longitudinal or transversal) patch stretching. To establish the properties of the materials used in the computational

model, we first empirically measured the tensile strength of dog bone-shaped patches of different formation (Fig. 3A) The 80% v/v GelMA − 20% v/v ACA combination demonstrated the highest stretchability (~110% yield strain) and elastic modulus (~20 kPa) and was therefore used in the computational models, which were then used to derive patch stiffness in the longitudinal and orthogonal directions sequentially. While many auxetic and non-auxetic architectures have been

described in the literature[27–29], we chose the commonly described re-entrant[12,30], sinusoidal ligaments[31], lozenge truss[32], rectangular truss[32] and honeycomb[33] architectures. Of note, these auxetic patch architectures have been explored in our previous work[17], but we needed to re-implement the design schemes and the associated computational modeling for the patch material to be able to derive the optimized patches for different dynamic organs. Here, we also explored some non-auxetic designs to be able to derive patches optimized for conformation to diaphragms and tendon tissue. We created a library of 64 designs demonstrating a broad range of auxetic, non-auxetic, and anisotropic properties by changing the width (w), height (h), radius (r), inclination angle (ia), and thickness (t), while keeping the overall patch size at $30 \times 30$ mm² (Fig. 3B, see Supplementary Table 1 for the values of the parameters for each design iteration of the patches). Using this systematic approach, the patch stiffnesses and Poisson's ratios were determined in the longitudinal and transverse directions (select computational outcomes are shown in Fig. 3C), as summarized in Fig. 3D, to map the different patch architectures to the dynamic loading of different organs. We also validated whether these patch architectures could withstand the different deformations of different organs by plotting the yield strains of these patches. Figure 3D shows that the yield strains of the patch formulations for different organs were always in excess of the maximum organ deformation. This approach allows the patches to retain their elasticity throughout repetitive cycles of organ deformation.

By considering the stiffness and Poisson's ratios of the different organs, as well as ensuring that the patches surpass the maximum organ deformation, we have developed a patch architecture and its iterations for each organ (Fig. 3D and Fig. 4A). Notably, all these patches exhibit remarkable ultra-stretchability (Fig. 4B) To validate our computational predictions, we proceeded with tensile testing of the selected patch geometries for different organs. The results revealed a close correlation between the computational estimates and the experimental outcomes (Fig. 4C). The experimental yield strains of the patches were slightly higher compared to the in silico predictions. This minor deviation observed is likely due to the fact that the computational models employed perfect geometries with sharp edges, which can give rise to concentrated stress. Conversely, the experimental patches featured sharp edges with a fillet radius due to process limitations, which likely reduced internal stresses.

We next tested whether the patches could conform to a dynamic organ by determining the change in organ surface area. Since each organ has a distinct shape that further varies throughout the organ, we used volumetric fold change of the organ as the rational criterion for determining the change in organ surface area (Supplementary Fig. 2A). Of the different organs, the human stomach undergoes the highest dilation (up to 2.2-times its original volume)[1], so we used the stomach as the exemplar by developing a mechanized setup to inflate and deflate a stomach-mimicking balloon (Supplementary Fig. 2B). The air flow rates inside and outside of the balloon were finely calibrated such that the expansion of the balloon mimicked the surface area change of the stomach and attached the selected stomach-mimicking Re-en itr1 patches onto the balloon model and studying the patch mechanics. Upon increasing the balloon volume to approximately 2.2-times its original volume (45 to 100 cm³) using cyclical pressurization through air, we demonstrated that the auxetic Re-en itr1 patches easily conformed to balloon expansion and contraction (see Supplementary Movie 4) and had a similar expansion ratio to that of the auxetic surface of the balloon. In contrast, a non-auxetic patch (Rect itr5, which is designed to mimic the mechanics of the muscular portion of the diaphragm, see Fig. 4A) did not conform to the balloon expansion and demonstrated a significantly smaller expansion ratio. These experiments indicate the advantage of the auxetic patches over a non-auxetic patch in applications concerning volumetrically deforming dynamic organs (bladder, stomach, heart, intestines, lungs). However, we would

like to note that the balloon surface is not the same as serosal surface of the organs and this model is also unable to exhibit the intrinsic spatial and directional anisotropy of the organs[1]. Therefore, to further ascertain the functionality of the patches, we used ex vivo lung models and in vivo heart models in the next study.

## AuxES patches demonstrate conformation to different dynamic organs

An important criterion for any dynamic organ patch is the need to withstand the cyclical stretching and contraction and the resulting fatigue loading after administration. We, therefore, used ex vivo lung (rodent and porcine) and in vivo heart (rodent) models to test patch compliance. For the rodent models, we scaled-down the patches by a scaling factor of s = 0.5 (i.e., $15 \times 15$ mm² patches with proportionally reduced features) and in porcine models we used a scaling factor s = 1.5 (i.e., $45 \times 45$ mm² patches). Herein, the scaling of the patches will not affect the stiffness and Poisson's ratios of the patches[1], since all the features are uniformly scaled, and their relative dimensions (e.g., the width and height ratios, etc.) remain the same. Unlike non-auxetic patches, the auxetic patches demonstrated a greater increase in overall surface area during physiological ventilation (PV) and hyperventilation (HV) states (Fig. 4D, E, $p < 0.05$) in both the rodent and porcine models (also see Supplementary Movie 5). Next, we applied the patches over the beating heart within an in vivo rodent model. The patches easily complied with the rapid motion of the heart and demonstrated rhythmic stretching and contraction in synchrony with the diastolic and systolic cycles of the heart (Fig. 4F, also see Supplementary Movie 6).

## Drug-loaded patches designed with high flexibility and stretchability for wound healing

Compared with dynamic organs such as the lung and the heart, the skin undergoes complex deformations during normal physiology. For example, the sole and the dorsum undergo the highest deformation during dorsiflexion and plantarflexion, respectively, and the Poisson's ratio is negative or positive across different regions[34]. Such a large variation in skin mechanics mandates the use of region-specific auxetic or non-auxetic patch architectures. Figure 5A demonstrates the high variability in the Poisson's ratios in different skin regions, in a synthetic anatomical foot model, from plantarflexion to dorsiflexion of the sole and the dorsum[34]. In some regions, the Poisson's ratio can be close to −0.6 (central region in the heel portion of the sole and toe portion of the dorsum), while other regions of the foot undergo minimal flexing. Thus, auxetic patches will be required to allow for flexible movement of the feet. Figure 5B demonstrates conformation tests of the patches in the toe regions of the sole (Re-en itr1 most suitable) and dorsum (Re-en itr4 most suitable) during dorsiflexion and plantarflexion in an anatomical foot model. The patches demonstrated excellent conformation to the movement of the foot (see Supplementary Movie 7); for example, on the sole, Re-en itr1 expanded along both the length and width during dorsiflexion, and contraction along both the length and width during plantarflexion.

After establishing the auxetic lattice design suitable for skin applications we next optimized drug loading onto the patches to promote wound healing. In our design, we specifically chose CNPs over other UV absorptive agents (such as FD&C yellow food dye[24]), as CNPs have added therapeutic benefits through their intrinsic anti-inflammatory[35], anti-microbial[36], anti-oxidant[35], and free-radical-scavenging[37] properties. We further hypothesized that we could exploit the intrinsic bioadhesive characteristic of the AuxES patches coated with biological therapeutics to further enhance their therapeutic potential. For this, we selected mesenchymal stem cell-derived extracellular vesicles (MSC-EVs) as the biological therapeutic[38–40], since MSC-EVs have been shown to have intrinsic regenerative effects by inhibiting inflammation[41,42], and promoting cell proliferation[43,44],

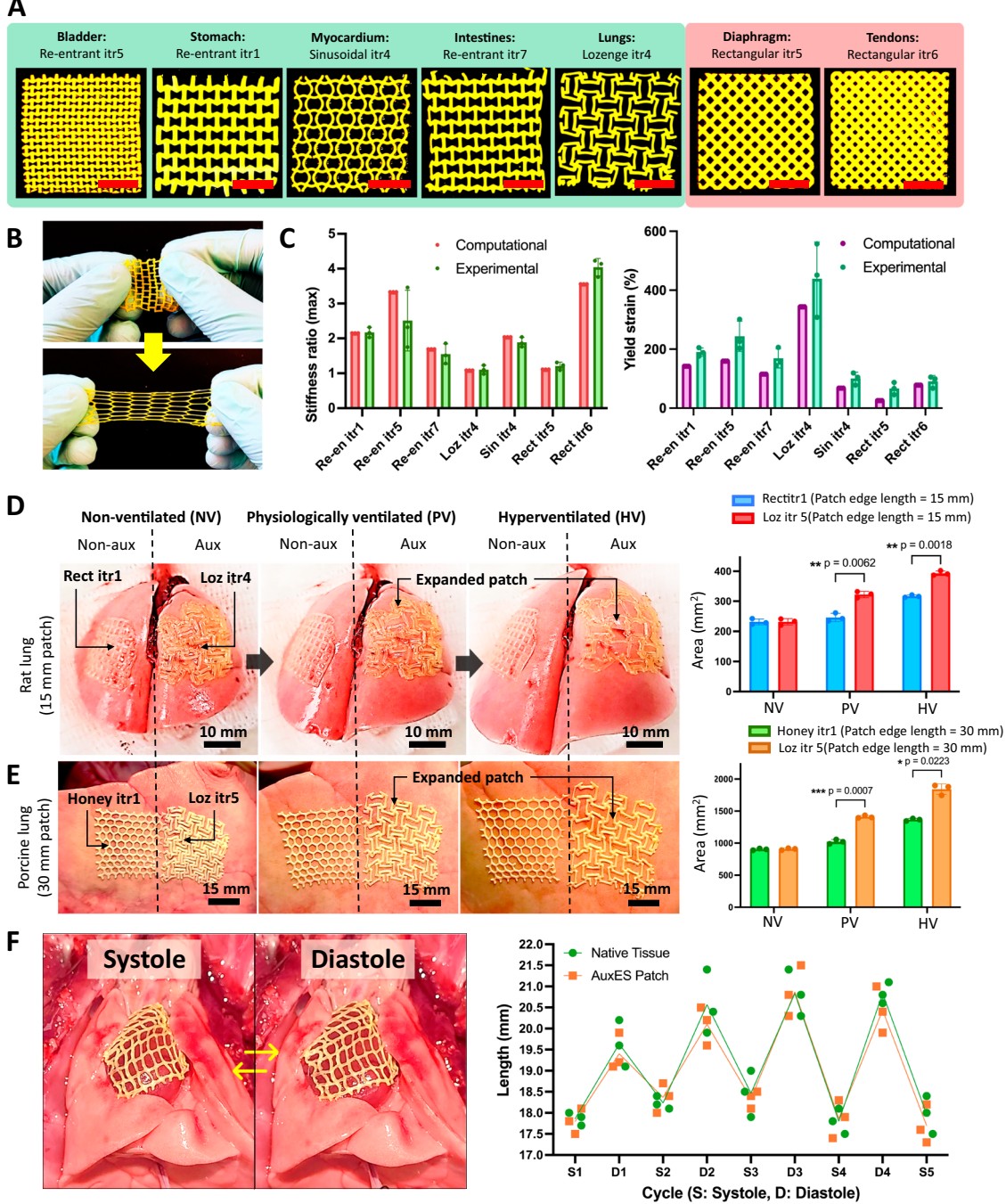

**Fig. 4 | Selected patches validate computational estimates and demonstrate conformation to dynamic organs. A** Printed patches based on the selected patch designs for different dynamic organs. Note that some are auxetic and some are non-auxetic, depending on the target organ. **B** Demonstration of patch ultra-elasticity (see Supplementary Movie 3). **C** Tensile testing of the selected patches for the different organs demonstrates a close correlation between the computational and experimental outcomes of the elastic modulus (stiffness) and the yield strains of the patches. **D** Ex vivo implantation of the patches (15) over rat lungs, and comparison of a patch with square holes vs an auxetic patch (Loz itr5). Auxetic patches substantially increased their surface area by increasing the size of the voids, thereby conforming to the lung mechanics. **E** Ex vivo implantation of the patches

(30×30 mm²) over the right middle lobe of porcine lungs, where auxetic patches (Loz itr4) better conformed to the lung mechanics compared with non-auxetic (honeycomb) patches (see Supplementary Movie 5). **F** Patches were implanted in vivo over beating rat hearts (see Supplementary Movie 6) The attached patches conform to the heart beat over several systolic and diastolic cycles. Lengths of patches across different cycles were measured via analysis of video frames in ImageJ.) Statistical significance in **D** and **E** was analyzed by two-way ANOVA followed by Šidák's multiple comparison tests,*$p < 0.05$, ** $p < 0.01$, ***$p < 0.001$. Data in **C**–**F** are from $n = 3$ biologically independent samples each and are presented as mean ± s.d.

potentially making them an effective treatment for wound healing. We first assessed whether MSC-EVs could be successfully attached to the patches by dipping the patches into a solution containing fluorescently-labeled MSC-EVs ($10^{11}$ MSC-EVs/ml), followed by

quantification. Our AuxES patches with active bioadhesion demonstrated higher EV loading capacity compared to pure GelMA patches (Fig. 5C) After 30 min in the EV-rich solution, GelMA patches only fluoresced at the edges due to negligible bioadhesiveness, while AuxES

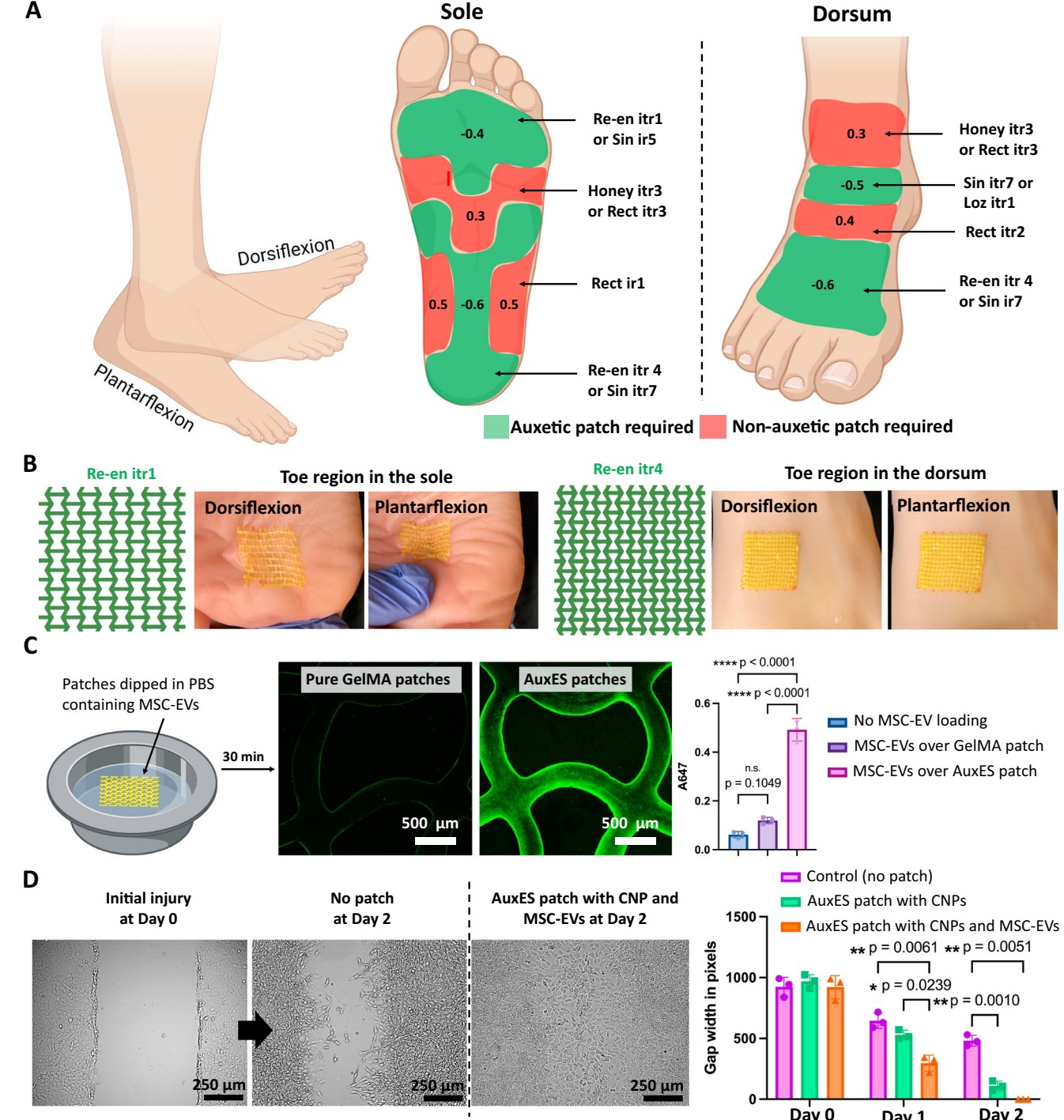

**Fig. 5 | Adaptation of AuxES patch confirmation to different skin regions in a synthetic skin model. A** There is high variability in patch deformation under different physiological motions of the skin. Taking the sole and dorsum as example regions, the Poisson's ratios of various regions during dorsiflexion or plantarflexion are indicated. The printed AuxES patches relevant to each sub-region are highlighted. Panel **A** was, in part, created with Biorender.com. **B** Patch deformation relevant to the different subregions of the sole or dorsum in an anatomical foot model under dorsiflexion or plantarflexion. The patches can conform to the skin dynamics in each case (see Supplementary Movie 7). **C** Due to their intrinsic bioadhesiveness, AuxES patches allow the attachment of fluorescently labeled MSC-EVs to their surface. After 1 h of incubation in an EV-rich solution, EV attachment to the AuxES patches is substantially greater than that of pure GelMA patches.

Statistical significance is analyzed by one-way ANOVA followed by Tukey's multiple comparison tests,*$p < 0.05$, ** $p < 0.01$,***$p < 0.001$, and ****$p < 0.0001$. Panel **C** was, in part, created with Biorender.com. **D** In vitro scratch assays with 3T3 fibroblasts demonstrate that patches laden with both CNPs and MSC-EVs clearly outperform patches laden with CNPs and the control groups (no patches). While the control groups only demonstrated ~40% wound healing after 48 h (days 0 to 2), CNP-laden patches demonstrated up to 90% healing, and the MSC-EVs and CNP-laden patches demonstrated complete wound healing. Statistical significance is analyzed by two-way ANOVA followed by Tukey's multiple comparison tests,*$p < 0.05$, ** $p < 0.01$. Data in **C** and **D** are from $n = 3$ biologically independent samples and are presented as mean ± s.d.

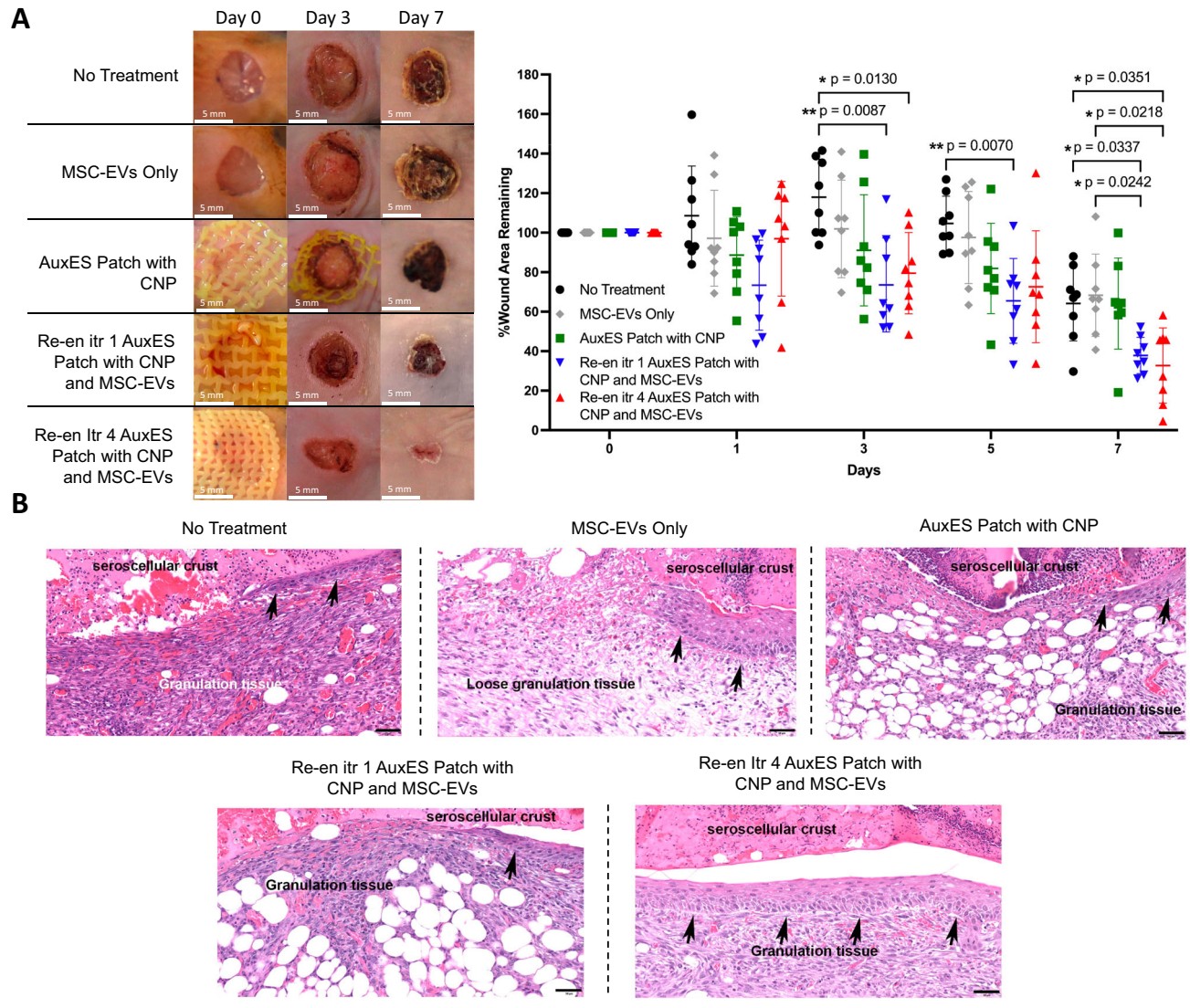

**Fig. 6 | *AuxES patches can be an effective treatment for wound healing*. A** Re-en itr1 and Re-en itr4 AuxES patches laden with CNPs and MSC-EVs demonstrated significantly greater wound closure compared with other treatment groups after a week of administration over cutaneous wounds in mice. Data are from *n* = 8 biologically independent animals and presented as mean ± s.d. Statistical significance was analyzed by two-way ANOVA followed by Tukey's multiple comparison tests,

*$p$ < 0.05 and ** $p$ < 0.01. **B** Photomicrographs of the margin of skin wound. Re-epithelialization (represented by the arrow (↑)) and granulation tissue are indicated in the figure. Hematoxylin and eosin (H&E) staining, 20x objective (scale bar = 50 μm). Staining was performed on all eight biologically independent samples per group over two independent experiments.

patches demonstrated MSC-EV coating across the entire patch surface. Furthermore, after loading, the EVs demonstrate a burst release in the initial 12 h for immediate therapeutic exposure, followed by sustained release over a week in culture (Supplementary Fig. 3).

We next used in vitro scratch assays[45] to evaluate the effectiveness of AuxES patches for facilitating wound healing. AuxES patches laden with both CNPs and MSC-EVs completely filled the scratch over 48 h, patches laden with only CNP resulted in ~90% filling of the scratch, while control groups without any patch treatment only filled 40% of the scratches after 48 h (Fig. 5D).

**AuxES patches are effective at wound healing and promote wound healing response**

The high conformation of AuxES patches to skin movements should prevent constrained movement after application and patch-induced mechanical stretch of the skin, lending themselves to the personalized treatment of pathologies such as burn wounds and diabetic ulcers with specific patches in different skin regions. We next assessed the wound

healing effectiveness of a week-long (i.e., day 0 to day 7) administration of the three groups of patches in a mouse model of cutaneous injury (Fig. 6A). We chose the Re-en itr1 and Re-en itr4 architectures as they were best suited for conforming to the skin movement on the foot dorsum and sole, as shown in Fig. 5A. The patches attached easily to the mouse backs and conformed to the movement of the animal. After a week, Re-en itr1 and Re-en itr4 AuxES patches with CNPs and MSC-EVs demonstrated significantly increased wound closure compared with all other groups, including the MSC-EVs Only group ($p$ < 0.05) indicating that there is improved wound healing when the AuxES patches are coupled with MSC-EVs. Hematoxylin and eosin staining of the periphery of the wounds indicated that the granulation tissue was mostly mild to moderate, meaning there was ample and appropriately modeled granulation tissue for the stage of the wound healing process. Importantly, there was no evidence of a foreign body response to the AuxES patch material (Fig. 6B).

MSC-EVs have been shown to promote cutaneous wound healing through macrophage polarization to the anti-inflammatory

M2 phenotype[46] and by promoting collagen secretion and angiogenesis[47,48], which likely synergized with CNPs to provide robust wound healing in the AuxES patches containing both the therapeutics.

Next, we sought to understand the biological response in the cutaneous wound as a result of the AuxES patch treatment, the presence of CNP, and the presence of MSC-EVs. For this, we included the following groups: AuxES patches without CNPs, AuxES patches with CNPs, AuxES patches containing both CNPs and MSC-EVs, or no patch. We employed Nanostring technology, which is an amplification-free technology that measures gene expression by counting mRNA molecules directly. We studied the mRNA expression in wound tissues. Only genes that were at least 2-fold up or downregulated were considered biologically relevant. Pathway analysis on the genes in the AuxES patches containing only CNPs and with both CNPs and MSC-EVs showed that the patches were promoting a wound healing and immunological response. Because dermal wound healing occurs typically in phases, we stratified the genes into inflammation, proliferation, and remodeling[49] to assess the therapeutic effect of patch treatment (Fig. 7A) The genes identified in this study were stratified via bioinformatic protein pathway software (Pantherdb and Reactome) as well as literature sources[50–57]. As visualized in Fig. 7A, the majority of the up and down regulated sequences fell into the inflammation phase of wound-healing. Of all the markers assessed in treated groups, only the markers from mice treated with the AuxES patches containing both CNPs and MSC-EVs showed greater than 3-fold upregulation (Fig. 7B) The most significantly upregulated inflammatory proteins within the AuxES patches containing both CNPs and MSC-EVs were *Ccr4, Ackr4, Cd7, Il-17b* and *Il-2*. The majority of these genes have been implicated in promoting the transition from an inflammatory to a proliferative environment. For example, the upregulation of *Ccr4* (*Cd194*, a receptor associated with chemotaxis, such as *Mcp-1* and *Rantes*) has been shown to promote wound healing through the recruitment of T-regulatory cells to the wound site, and is thus a precursor to the proliferation phase of wound healing[58]. *Ackr4* has been found to be critical in promoting the egress of dendritic cells, as well as in the removal of inflammatory cytokines[59]. The *Il-17* family has been identified as key mediators of leukocyte infiltration and T-helper cell-mediated inflammation[60]. Finally, within diabetes and lupus wound models, dermal wounds did not heal in IL-2 knockout models. Furthermore, where *Il-2* was present, the skin regrowth was stronger, and T-cell expansion was more highly regulated[61]. Furthermore, we observed that within the later wound-healing phases indicated as proliferative and remodeling, the AuxES patches containing both CNPs and MSC-EVs had nearly a third of the proteins that were upregulated, and this is visualized in Fig. 7A. Pathway analysis of these upregulated proteins show primarily immune-cell activation and interleukin signaling (Fig. 7C, D)

After further analysis, we discovered that the AuxES patches were upregulating both proliferative and remodeling wound healing phase markers to a greater extent than any of the other treatments. This, in combination with the visual improvement in healing seen with the AuxES patches containing both CNPs and MSC-EVs, indicates that the patches are indeed promoting a more rapid wound healing response than the other treatments.

### Void-filled AuxES patches are an effective treatment of puncture wounds

While auxetic patches are well suited to many dynamic organ pathologies (e.g., myocardial infarction treatment[12,31]), they are unsuited for pathologies involving loss of bodily fluids[9] or air (pneumothorax[62]). We therefore modified the AuxES patches using a lower concentration GelMA-ACA mixture to create hole-filling, auxetic patches by filling the voids in the auxetic lattice within the patches (Fig. 8A). Using the same material to fill the voids can be a significantly simpler and cost-effective alternative to tissue glues (such as Fibrin) which are

commonly used as sealants. Optimization of the concentration of void-filling (VF) material to successfully fill puncture wounds while still retaining the auxetic properties is shown in the Supplemental information (Supplementary Table 2 and Supplementary Fig. 4). In this formulation, the 80% v/v GelMA – 20% v/v ACA mixture used for the patch lattices was further diluted to 50, 60, 70, 80, and 90% of its original concentration by mixing with PBS and application to the patch voids followed by UV exposure. Void-filling material diluted to 80% of its original concentration could allow the patch to retain its auxetic properties (see Supplementary Movie 8 demonstration of the conformation of VF-AuxES patches to the deformation of a balloon; Fig. 8B shows the balloon dilation measurements of the optimized VF-AuxES patches), while maintaining its integrity during expansion. This combination of void-filling materials into patches could be an effective strategy for the treatment of pulmonary air leakage (Fig. 8C).

We further explored the use of VF-AuxES patches (Loz itr4 geometry) for restricting pulmonary air leakage in an in vivo rat model. Pulmonary air leakage was induced in SD rats by open chest surgery and puncturing the lungs with an 18 G needle. The instantaneous and strong bioadhesiveness of the patches allowed patch placement while the lung underwent normal physiological motion (Fig. 8D, also see video of VF-AuxES patch over lung in Supplementary Movie 9), a clear advantage over conventional patches requiring the lung to be static for application[63]. We compared the change in tidal ventilation pressure in injured lungs without treatment or after treatment with simple AuxES patches (Loz itr4, without void-filling material) and VF-AuxES patches. We did not include a non-auxetic patch or a VF-non-auxetic patch as a control, as we had already established in the ex vivo rat models (Fig. 4D) that the auxetic patches conform to lung mechanics better than a non-auxetic patch. Healthy lungs demonstrated a ventilation pressure of 7.2 mm $H_2O$ under dilation, which was reduced to 5.8 mm $H_2O$ in injured lungs (Fig. 8E). As expected, there was no improvement in lung function in the simple AuxES patch group, but VF-AuxES patches fully restored lung function (Fig. 8E).

## Discussion

There has been a steady rise in the development and use of therapeutic patches[3,64–67] over the last few decades for a wide range of applications including myocardial infarction[68], chronic wounds[69], organ hemorrhage[8,9], and cancer[70,71]. However, most of these patches do not readily adhere to wet and dry tissues and do not consider the complex mechanics and volumetric changes of dynamic organs caused by their auxetic and anisotropic properties. Here, we present an AuxES patch composition and design that addresses the limitations of conventional patches. Our AuxES patches exhibit properties which allow their easy adaptation towards dynamic organs: 1. ultra-elasticity (stretchable up to 400% its length without breaking); 2. instantaneous bioadhesion to dry and wet tissue (strong bonds form in <1 s over wet tissues) with detachment forces are higher than patches applied via commercial fibrin glue); 4. cost-effective, off-the-shelf components which are compatible with photo-projection based printing; 5. wound healing capability in vivo (through the controlled delivery of CNPs and conjugated MSC-EVs); and 6. possible synthesis with void filling to heal puncture wounds.

Notably, our material composition features yield strain up to 100% and when combined with the auxetic architectures, it exhibits yield strains up to 400%. This is a substantial increase in range when compared to published patches and materials[72,73]. Furthermore, we demonstrate a broad range of elastic moduli (1–20 kPa), stiffness ratios (1.0 to 4.0) and Poisson's ratios (−0.9 to 0.2), which allows our patches to cater to a broad portfolio of dynamic organs.

The defined (375 μm) resolution derived through the optimized CNP concentration allowed the printing of patches that closely mimicked their intended designs, and closely recapitulated the simulated mechanical properties in tensile tests (Fig. 4C). Resolution could

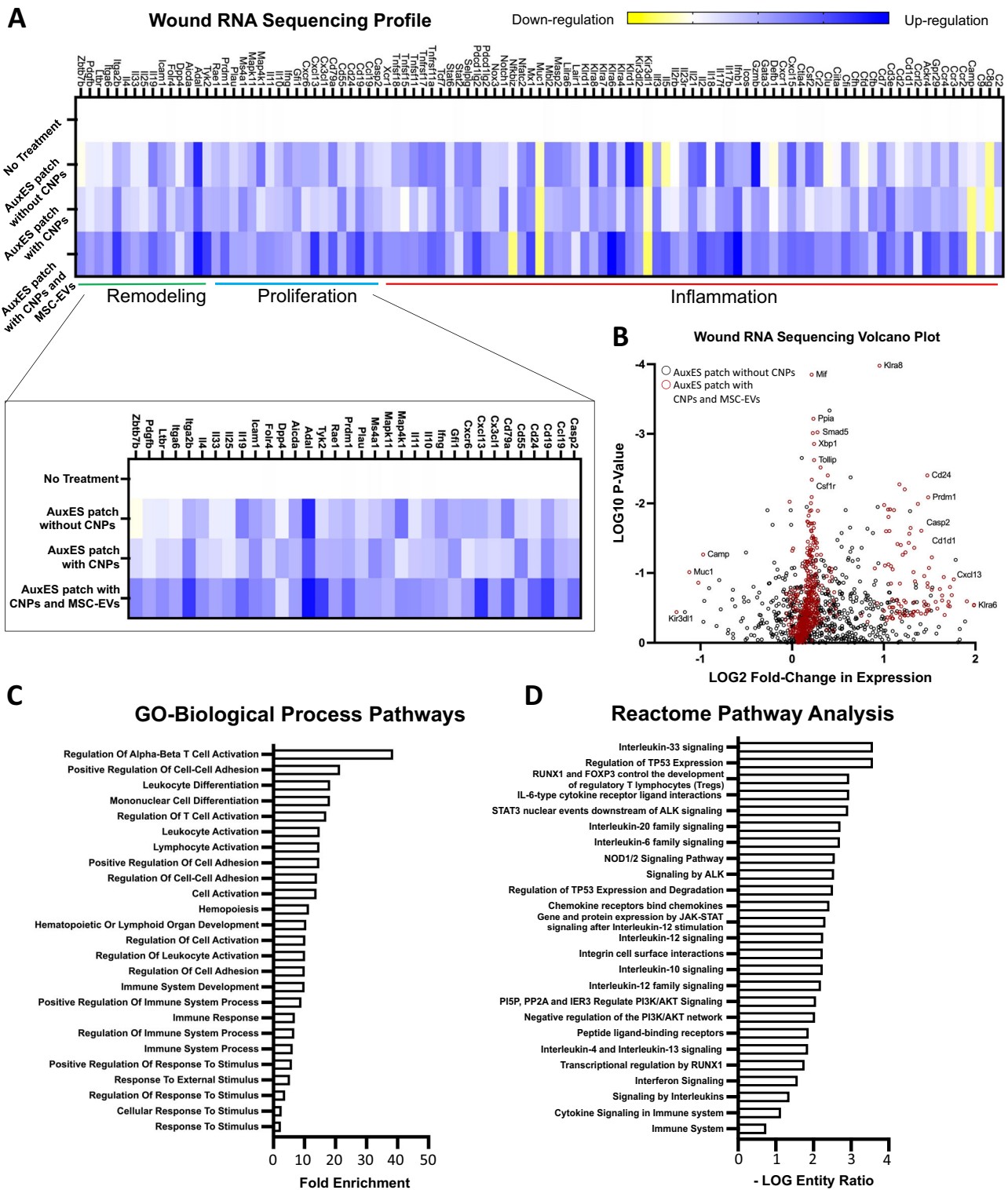

**Fig. 7 | AuxES patches loaded with CNP and MSC-EVs upregulate genes associated with proliferative and remodeling stages of wound healing. A** Heatmap depicting expression changes normalized to the no-patch treatment group, with expression levels equaling No Patch colored white. Upregulated genes are depicted in dark blue, down-regulated genes highlighted in yellow. Zoomed in heatmap from panel (a) highlighting the genes annotated as mediators in the proliferative or remodeling phases of wound healing. Identical color scale to larger heatmap. Data are from $n = 3$ biologically independent samples per group and are presented as gene fold change over no treatment group. **B** Volcano plot of differential gene expression, with the black points indicating the AuxES patch without CNPs

compared to the No Treatment-patch group, and red as AuxES patch with both CNPs and MSC-EVs compared to no patch. The Y-axis is the negative log10 of the p-value as calculated with multiple two tail t-tests for each gene. The X-axis is the log2 fold-change in expression of a given gene. Variance assumptions assumed individual variances for each row, and multiple comparisons were tabulated using False Discovery Rate (FDR), and the two-stage step up method (Benjamini, Krieger, and Yekutieli) with desired Q of 1.00%. **C** Gene ontology (GO) biological process pathways expressed as fold enrichment as compared to mus musculus gene list. **D** Reactome pathway analysis depicted with the -LOG of the entity's ratio, or the ratio of genes provided vs the total pathway components.

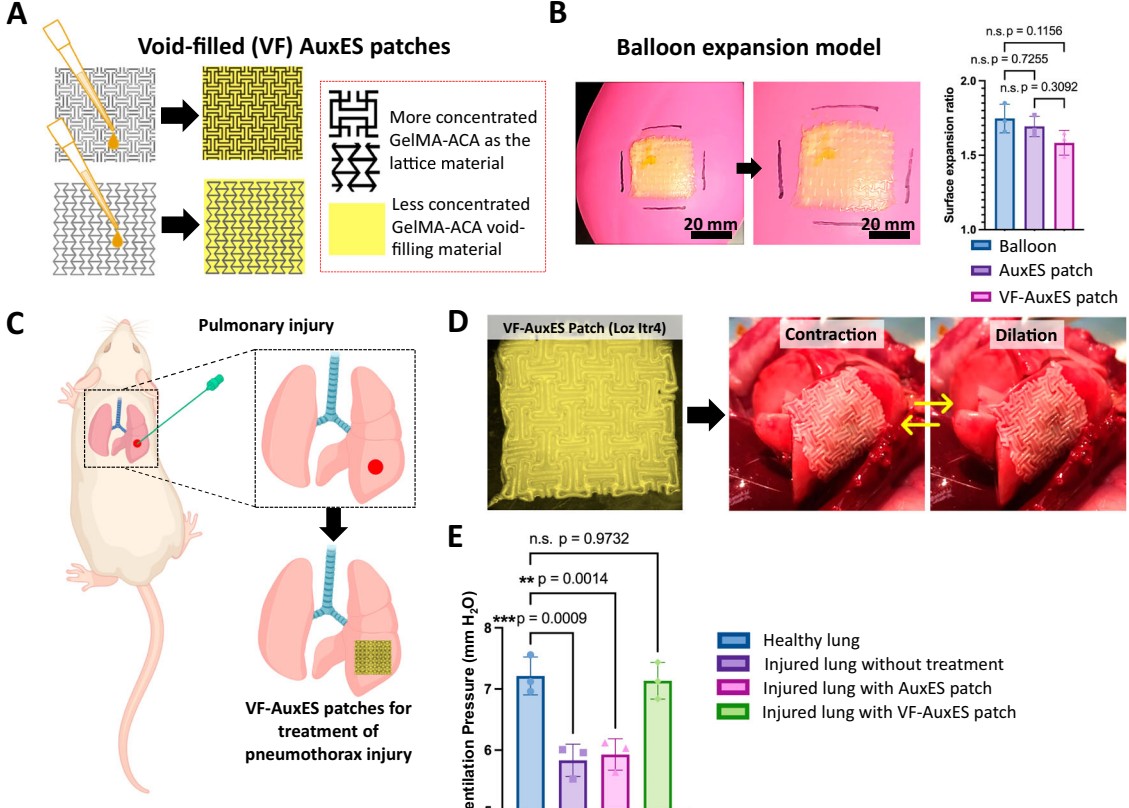

**Fig. 8 | Void-filled (VF) AuxES patches for the treatment of puncture injuries.**
**A** VF-AuxES patches were fabricated by filling the voids of AuxES patches using a reduced concentration of matrix containing GelMA and ACA. **B** AuxES patches synthesized using a 20% lower concentration of material than that used for the lattices (80% v/v GelMA and 20% v/v ACA solutions) demonstrated auxetic properties on balloon deformation without opening matrix pores (also see Supplementary Movie 8). **C** Concept of using VF-AuxES patches for the treatment of pneumothorax in an in vivo rat model. Panel **C** was in part created with Biorender.com. **D** Lung-mimicking VF-AuxES patch and its placement over rat lung (also see Supplementary Movie 9). **E** After inducing pulmonary injury, the tidal ventilation pressure during inspiration of the lung reduced from 7.2 to 5.8 mm $H_2O$. VF-AuxES patches fully restored the ventilation pressure to physiological levels, while simple AuxES patches were unable to improve lung ventilation due to air leaks through the empty voids within the patch lattices. Data in **B** and **E** are from $n = 3$ biologically independent samples/animals and are presented as mean ± s.d. Statistical significance was analyzed by one-way ANOVA followed by Tukey's multiple comparison tests in **B** and Dunnett's multiple comparison tests in **E** *$p < 0.05$, ** $p < 0.01$, and ***$p < 0.001$.

be further improved by reducing the voxel size of the incident light beam[23]. While the current system used chain-growth polymerization of the GelMA matrix in the presence of LAP, future research could use step-growth polymerization using thiol-ene photoclick chemistry[74,75] to allow for both higher resolution fabrication and quicker fabrication times. One could also deploy higher resolution techniques such as volumetric printing[76,77] or stereolithography[78] to achieve resolutions of up to 50 μm. Recently, volumetric printing techniques capable of rapidly printing multimaterial constructs have also been demonstrated[79]. Techniques such as filamented light (FLight) fabrication[80] could also be used to achieve aligned cell-scale (5−30 μm) microstructures within the patches which allow better cell infiltration. Further, process hybridization[18] can also be a powerful technique to circumvent the limitations of light-based fabrication through appropriately combining other techniques deploying sound[81,82] or magnetic field[83,84] to pattern reinforcing biomaterials within photocrosslinked constructs. Since the patches can be manufactured in a single layer, custom masks can also be used with UV lamps[81] to fabricate the patches, negating the need for complex projection systems in 3D printers and further reducing the cost of fabrication.

The patch composition is simple enough to be replicated and fabricated, and it can be linearly scaled-up for large-scale patch fabrication. Moreover, the AuxES patches are one of the few that instantly and strongly adhere to wet tissue, as shown in Figs. 2 and 4. Conventional GelMA does not show any tissue adhesiveness, only cell

adhesion which is intrinsic to all GelMA-based matrices due to the presence of RGD moieties[22].

These properties are important for future clinical translation of the technology since organs do not need to be immobilized for several minutes for their application, a current requirement for many patches reliant on the formation of amide bonds for crosslinking[8,9]. While other mussel-inspired formulations relying on hydrogen bonding[85] and free radical polymerization[86] have demonstrated instantaneous adhesion, they rely on dopamine and its catecholic groups, which frequently undergo oxidation in neutral and basic conditions and require a fine redox balance to maintain tissue adhesion. This is difficult to achieve and severely compromises their adhesion and limit their practical applications in medicine[87]. Further, our rational design framework for organ-specific soft anisotropic-auxetic patches ensures fatigue resistance over long-term administration while preventing undue stress on the organ.

Our wound healing experiments strongly suggest that a dual-therapeutic patch system containing CNPs and MSC-EVs is highly effective for promoting wound healing. Notably, AuxES patches demonstrated gradual dissolution into the skin after a week (Fig. 6A) and do not need to be removed. A degradable patch is desirable as it eliminates the need for dressing changes, which could lead to bleeding, skin tearing, and other related injuries. However, a fast degradation may be less advantageous for deep-scar wounds or chronic injury such as myocardial infarcts[88]. Here, the disintegration rate can be fine-tuned

and controlled through photo-crosslinking density[20] and thus tailored to different types of therapeutic application. A higher density of covalent crosslinks within the patches could be used to prolong the time to dissolution and could be fine-tuned for different applications by increasing the degree of methacrylation and concentration of GelMA or the content of photoinitiator and UV exposure[20]. The patch system containing CNPs and MSC-EVs was highly effective at promoting a wound-healing response, as shown by the studies in Fig. 6. Dermal wounds go through a four-stage process of wound-healing; (1) hemostasis (~0-12 h after injury), (2) inflammation, (3) proliferation, and (4) remodeling. Because the injury model presented in this work extends to > 5 days, hemostasis is not under consideration as a major component of the observed wound healing. The inflammatory phase takes place anywhere from 1-8 days after injury, and is demarcated via expression of interferons, leukocyte migration, activation of M1 macrophages, natural killer cells, and dendritic cell infiltration[50–55]. While chronic inflammation can lead to scarring and formation of fibrotic tissue because the time course of our injury models is less than ten days, the inflammatory markers observed are that of a normal wound healing process. The proliferation phase can begin as early as day 1, and lasts up to 8 days post injury[55–57]. The treatments containing both CNPs and MSC-EVs had significantly more up-regulation of proliferative genes than the other two treatments, which we hypothesized is indicative of a more advanced stage of wound healing. Furthermore, the treatments with CNPs and MSC-EVs also had a higher expression of remodeling phase markers (Fig. 7) The remodeling phase begins on day 3 and can extend for months after the injury, so an upregulation in these markers is indicative of a more advanced stage of wound healing[55–57]. For future applications of these AuxES patches where CNPs may not be needed, the patches can also be fabricated using food dyes (such as Sunset Yellow FCF) as the photo-absorptive agent[24]. Notably, our AuxES patches exhibit adhesive properties on both sides, which can be advantageous in certain applications. However, when using these patches for internal organs, it is important to consider the potential risk of the patch surface adhering to unintended organs or blood vessels, which could lead to further injury in the surrounding regions. To address this concern, we have conducted additional experiments (Supplementary Fig. 5), focusing on the development of a non-fouling coating for the patches while preserving their mechanical functionality. In these experiments, we coated the top part of the patches with polyethylene glycol diacrylate (PEGDA), which serves as a non-adhesive layer. To facilitate differentiation and imaging of the PEGDA and AuxES patch layers, we utilized rhodamine and fluorescein isothiocyanate (FITC) labeling, respectively. Our findings clearly demonstrate that the patches exhibit adhesion exclusively through the AuxES patch layer, while the PEGDA layer shows negligible adhesion.

While our AuxES patches have demonstrated adhesive characteristics when exposed to blood (Supplementary Movie 10), long-term incubation in blood may potentially affect their adhesion properties due to patch swelling. For instance, our current AuxES hydrogel-based patches can swell up to 200% in size compared to their original size ($10\times10$ mm$^2$) after 24 h incubation in blood at 37 °C (Supplementary Fig. 6). We have successfully demonstrated that this swelling can be controlled through post-curing the patches. Post-curing involves washing the fabricated patches, followed by incubation in water containing the photoinitiator, and subsequent exposure to UV light to enhance crosslinking of the patch matrix. Through this approach, we were able to reduce patch swelling in blood to a maximum of 50% increase in size compared to their original size (Supplementary Fig. 6).

Only a few patches have been clinically approved so far[89,90], highlighting the challenges to clinical effectiveness and translation. Other adhesive materials have also been investigated with promising strong and instantaneous adhesion[91] and ultra-elasticity[92], but the materials lack rationally designed anisotropic and auxetic architecture, ultra-elasticity, instant bioadhesion, and compatibility with high

resolution printing techniques. Regardless of the material used, our rational design framework demonstrated in this work can be used to design the anisotropic and auxetic patches to conform to specific organs, thus improving long-term integration of patches within hosts. With our VF-AuxES patches, we have demonstrated how this rational design framework can be further expanded to also include the treatment of dynamic wounds such as pulmonary air leakage. The VF-AuxES patches retained their auxetic properties whilst also preventing gas leakage from the lung. A void-filling patch system can potentially also be used for the treatment of bleeding wounds[93,94], such as those inflicted during hemorrhage. Notably, the materials may need fine-tuning for allowing patch adhesion over a continuously bleeding site. As for the auxetic patches with open holes, potential applications include the delivery of therapeutics for diabetic wound healing[95,96], burn wounds, treatment of myocardial infarction[12,68] or stomach ulcers[97,98]. Our future studies will investigate the use of these patches laden with or without therapeutic substances (e.g., small molecules, larger proteins or EVs) for treating these pathologies. Additionally, we plan to investigate whether the crosslinking mechanics and material constitution of the patches can be fine-tuned to match the actual stiffness of the organs, where the patches can be used to provide mechanical support to the organ (e.g. cardiac patches). This is especially important to reduce wall stresses at the infarct and border zones in the heart, following an ischemia reperfusion injury[68,99].

In conclusion, our patch technology provides a versatile platform for easy and cost-effective adaptation towards different dynamic organs, with potential applications across a broad range of pathologies including battlefield injuries, myocardial infarction, chronic wounds, pneumothorax and other diseases.

## Methods

### Regulatory
All the animal studies were approved by the Institutional Animal Care and Use Committee (approval ID: 21-268.0, 23-029.0, 20-045.0) at University of North Carolina at Chapel Hill.

### Patch materials synthesis
GelMA was prepared using existing protocols for the controlled methacrylation of gelatin[84]. Briefly, 5 g of gelatin (Bloom 300, porcine derived, Sigma Aldrich, St. Louis, MO) was dissolved at 10% w/v in 0.25 M carbonate-bicarbonate buffer[100] containing 2% w/v of sodium bicarbonate and 0.12% w/v of sodium carbonate in deionized water. The gelatin solution was kept at 50 °C until achieving a clear solution. Next, 159 µl of methacrylic anhydride (MA) was added and the reaction allowed to run for 60 min at 50 °C. The reaction was stopped, and excess MA removed, by the addition of 100 ml of 1:1 mix of pure ethanol and acetone. The degree of methacrylation was 40%, as determined by $^1$H NMR[100]. Some of the preliminary data was generated using GelMA synthesized by Rousselot Inc. Acrylic acid was used as purchased (79-10-7, Sigma Aldrich). CNPs were prepared using established methods[101,102]. Briefly, curcumin (Cur) powder (C1386, Millipore Sigma) was dissolved in tetrahydrofuran (THF) solution at 25 mg/mL. 50 µL of the Cur-THF solution was rapidly injected into 450 µL of deionized water with vigorous stirring at 1400 rpm to aggregate as nanoparticles. The CNP suspension was air-dried to remove the organic solvent and lyophilized. The resultant CNPs were stored at −20 °C until further use. To prepare the material for the patches, 10% w/v GelMA in PBS at 37 °C and acrylic acid (79-10-7, Millipore Sigma) were mixed at different ratios (80%–20% or 90%–10%, also see results in Fig. 3A), followed by the addition of 0.03% w/v LAP photoinitiator and the desired amount of CNPs (1–2.5 mg/ml) to formulate the photoink.

### Patch printing and postprocessing
The material was cooled to room temperature (24 °C), and 750 µl was added to a rectangular enclosure ($50 \times 50$ mm$^2$) attached onto a

PDMS-coated petri dish and uniformly distributed across the enclosure using a pipette tip. Acrylic acid prevented the thermo-reversible crosslinking of GelMA, which is otherwise a native property of GelMA hydrogels[22]. The petri dish was then placed above a digital light projection system (LumenX, Cellink AB) and projection printed using 405 nm UV light at 20 mW/cm² for 2 min. Patches were then neutralized and further crosslinked by dipping into a solution containing 0.5 M CaCl₂ and 0.05 M NaOH in deionized (DI) water. The patches were gently washed with DI water and allowed to air dry for up to 60 min within a biosafety cabinet to improve adhesiveness (Supplementary Fig. 1A). To fabricate the bilayered patch (Supplementary Fig. 5), we labeled PEGDA (5% w/v in DI water, with 0.03% w/v LAP) using rhodamine and the AuxES patch material using FITC[79]. First, AuxES patches were fabricated using the method described above. Additionally, a thin 200 μm coating of PEGDA was applied to a separate substrate, followed by carefully placing the AuxES patch over the thin layer, ensuring that the project image (in this case, Sin itr1 architecture) aligned perfectly with the AuxES patch. Post fabrication of the bilayered patches, the patches were washed and treated with CaCl₂ and NaOH, followed by air drying, before imaging in a confocal microscope (Olympus Fluoview 3000), or testing for bioadhesion. To assess swelling (upon incubation in blood, PBS, or water for 24 h at 37 °C), the post-curing of the patches was performed by washing the patches in DI water to remove non-photocrosslinked material, followed by incubating the patches in water containing 0.03% w/v LAP for 15 min. The patches were then exposed to bulk UV exposure (obtained by projecting an image comprising of a simple square 30 × 30 mm²) at 20 mW/cm² for 2 min. The patches were then re-washed with DI water and treated with 0.5 M CaCl₂ and 0.05 M NaOH in deionized (DI) water prior to the swelling experiments.

### Tensile testing
The tensile properties of the patches were determined using an in-house setup consisting of a linear stage actuator (101-80-124, SainSmart) as the stretching mechanism, and a 5 kg load cell (TAL220B, Sparkfun) and amplifier (HX711, Sparkfun) as the stationary anchor. The linear actuator was controlled via a stepper driver (101-60-197, SainSmart) and motion controller (101-60-199, SainSmart). The ends of the patch were clamped between the linear actuator and the load cell, and the patches stretched 0.125 mm/s similar to previous existing studies[103,104]. The load cell was connected to an Arduino Uno and provided the stress vs. strain curve to determine the mechanical properties.

### Bioadhesion measurements
Patch bioadhesion was measured over ex vivo mouse livers. Briefly, the mouse livers were gently dabbed using blotting paper to remove excess blood, followed by placement over the stationary load cell of the tensile testing apparatus. Next, patches were placed on the mouse liver. For GelMA patches administered using fibrin glue, the fibrin sealant was first prepared as per the manufacturer's guidelines (TISSEEL, Baxter). GelMA patches were placed on the mouse livers, followed by the placement of 50 μl each of the fibrinogen prepolymer solution and the thrombin crosslinker solutions to attach the patches to the mouse liver. All patches were administered such that one end of the patches hung over the edge of the liver by 5 mm, which was attached to the linear translation stage of the tensile testing apparatus. The patches were stretched at 0.125 mm/s, and the highest force generated during the stretching procedure was noted as the detachment force.

### Evaluation of patch biocompatibility in vitro
NIH 3T3 fibroblasts (CRL-1658, ATCC) were cultured in Dulbecco's modified Eagle's medium (DMEM) and 10% fetal bovine serum and 1% penicillin-streptomycin in six-well plates (2 ml of medium per well) until 40% confluency. The patches were then added into the wells and the viability assessed after 2 days using the Live/Dead™ assay (L3224, ThermoFisher Scientific).

### Patch designs
Solidworks (Dassault Systems) was used to prepare the patch designs. The inset images (Fig. 3B) constituted the repetitive elements in the linear array of the lattice. The dimensional parameters (h, w, r, t, te, and ia) were defined as global variables, so that they could be easily altered to change the overall patch design. The overall size of the patches, irrespective of the dimensionality of the lattice elements, was kept at 30 × 30 × 1 mm³.

### Computational modeling of the patches
The 3D patch geometry was imported into the structural mechanics module of COMSOL Multiphysics (COMSOL Inc.). A fixed boundary condition was applied on one edge of the patch (either the longitudinal (L, assumed along x-axis) or transverse (T, assumed along y-axis) direction) and a linear displacement of $\delta = 10\%$ strain applied on the opposite edge. A mesh density of 1/10th of the minimum element size (0.25 mm) was selected as previously[105]. The stiffness of the patch (E) was determined from the strain energy ($U_S$), strain ($\delta_L$ or $\delta_T = 10\%$), and volume of the patch ($V_P$) as per $E = \delta^2 V_P / 2U_S$. For some patches, this stiffness was different when stretched longitudinally (along x-axis), than when stretched transversally (along y-axis), thereby making the stiffness ratio smaller or larger than 1. The Poisson's ratio ($v_p$) was determined from the deformation transverse deformation when stretched longitudinally, hence $v_p = \delta_T / \delta_L$. The yield strain ($\delta_{Yield}$) of the material was calculated at the stretch when the maximum internal stress (von mises = $\sigma_i$) within the patches exceeded the yield strength ($\sigma_{max}$) of the bulk material: $\delta_{Yield} = \sigma_i / \sigma_{max}$.

### In-house balloon model to demonstrate patch compliance to organ mechanics
The balloon model consisted of a programmable control of a pneumatic valve using a linear actuator. An air flow meter was used to calibrate the air flow (VFA-26-BV, Dwyer Instruments). An open cylindrical tubing attached to the inlet of the balloon allowed the input air to exit the balloon. To measure the change in the surface area of the patches compared with that of the balloon, point marks were placed on the portion of the balloon encompassing the patches and around the patches. The relative expansion of the patch and the balloon was measured by calculating the distance between the opposite edges of the patches, and the opposite points placed along the balloon, in ImageJ. The product of the measured length (L) and width (W) was the surface area. The expansion ratio was the ratio between the change in surface area of the patches ($L_P \times W_P$) to the change in surface area of the square encapsulated by the points on the balloon ($L_B \times W_B$).

### Ex vivo evaluation of patch compliance
Rat lungs were excised from anesthetized male Sprague Dawley (SD) rats (~300 g average weight, 12 weeks old) after euthanasia. Porcine lungs (cold flushed post euthanasia) were obtained from North Carolina State University School of Veterinary Medicine (UNC Institutional Animal Care and Use Committee (IACUC) Protocol No: 20-045.0). Patches were attached to the lungs using forceps. The lungs were then ventilated using a rodent ventilator and a Servo-i mechanical ventilator, respectively. The ventilation volumes were 7.5 ml/kg (physiological ventilation) and 12.5 ml/kg (hyperventilation) for rat lungs, and patch motion was recorded as a video. The patch size was determined as for the balloon model.

### In vivo evaluation of patch compliance
12 week old, male and female SD rat were purchased from Charles River. An intraperitoneal injection of ketamine/xylazine was used to

anesthetize the SD rats and the rats (UNC IACUC Protocol No: 23-029.0) were kept unconscious while ventilating at a volume of 7.5 ml/kg (typical for rodent studies[106]) with isoflurane and oxygen (after tracheotomy). The lungs and the heart were exposed via sternotomy. To demonstrate the compatibility of the patches with minimally invasive surgery, the patches were wrapped around a minimally invasive surgical scope and attached onto the heart. Patch motion was recorded as a video. The changes in patch length were determined using analysis of different frames of the captured video in ImageJ.

## In vitro healing evaluation using scratch assays

Methods based on previous work[45] were used for the scratch assay. Briefly, 3T3 cell cultures within six-well plates were allowed to reach 100% confluency and were then scraped using a 23 G needle to create ~650 μm (wide) and 5 mm (long) scratches through the cells. The cells were then allowed to proliferate through the scratch without any treatment (control), or in the presence of patches (placed within the wells after creation of the scratch). The width of the scratch was determined after 24 h and 48 h using analysis of brightfield images in ImageJ.

## In vivo murine wound healing model

Female, 6-8 week old *C57BL/6J* were purchased from Jackson Laboratory. The animal studies were approved and carried out in compliance with the Institutional Animal Care and Use Committee standards (UNC IACUC Protocol No: 21-268.0). The mice were housed individually with 12 h light−dark cycles. For wounding, mice were anesthetized using gaseous isoflurane and received a subcutaneous injection of 0.05 mg/kg buprenorphine. Hair was removed from the dorsal region of the mouse using clippers and a depilatory cream, and the skin was prepared for surgery using betadine and 70% ethanol. A sterile 6 mm biopsy punch was used to outline a circular pattern between the shoulders. A surgical skin marker was used to mark the boundaries. Forceps were used to lift the skin, and surgical scissors were used to create a full thickness wound on each mouse. Mice were randomly assigned into five treatment groups of 8 mice each and treated with AuxES patches with CNPs, Re-en itr1 AuxES patches with CNPs and MSC-EVs, Re-en itr4 AuxES patches with CNPs and MSC-EVs, only MSC-EVs or no treatment. Patches were adhered to the wound site immediately following wounding. For the only MSC-EVs group, MSC-EVs in PBS were topically applied to the wound to serve as a control. After the initial treatment, wounds were covered with a Band-Aid to prevent patch removal by the mice in the first 24 h post-wounding and treatment. Every alternate day, all wounds were imaged and measured in perpendicular directions on ImageJ for wound area calculations. On day 7 post-wounding, mice were sacrificed, and residual wounds were harvested and immediately fixed in 10% neutral buffered formalin.

## Histopathology of in vivo skin wounds

At day 7 after wounding, the wound and surrounding skin were collected from all mice. Samples were fixed in 10% neutral buffered formalin for ~72 h and routinely processed in paraffin, embedded, and sectioned to 5 μm, stained with hematoxylin and eosin, and evaluated microscopically. Wounds were trimmed at approximately the middle of the wound bed.

## Pathway analyzes of NanoString RNA sequencing data - immune gene detection and quantification

Isolation of mRNA was performed as follows[107]. Briefly, cells were lysed with TRIZOL buffer (Sigma) and total RNA was isolated by chloroform extraction and quantified using a nanodrop 2000™ spectrophotometer. NanoString technology and the nCounter Mouse Immunology Panel was used to simultaneously evaluate 561 mRNAs in each sample. Each sample was run in triplicate. Briefly, a total of 100 ng mRNA was hybridized to report-capture probe pairs (CodeSets) at

65 °C for 18 h. After this solution-phase hybridization, the nCounter Prep Station was used to remove excess probe, align the probe/target complexes, and immobilize these complexes in the nCounter cartridge. The nCounter cartridge was then placed in a digital analyzer for image acquisition and data processing. Hundreds of thousands of color codes designating mRNA targets of interest were directly imaged on the surface of the cartridge. The expression level of each gene was measured by counting the number of times the color-coded barcode for that gene was detected, and the barcode counts tabulated. nSolver v4.0, an integrated analysis platform was used to generate appropriate data normalization as well as fold-changes, resulting ratios and differential expression. Replicates ($n = 3$) of sequence reads were normalized to the control (no patch) for downstream analysis. Sequences that were of identical expression levels in all samples were excluded from pathway analysis as background. Sequences in the groups AuxES patch containing only CNPs, or AuxES patch containing both CNPs and MSC-EVs that were calculated as being 2-fold higher or lower compared to the no treatment group, were used for pathway analyzes. These fold-changes in expression were analyzed using GraphPad Prism and statistics conducted using a two-way ANOVA for grouped analyzes. Up- or downregulated sequences were analyzed using the Panther protein database overrepresentation test, under the GO-Biological process complete analysis, with Fisher's Exact test and corrections via calculation of false-discovery rate.

## MSC-derived extracellular vesicles (MSC-EVs) isolation, fluorescent labeling, and attachment to the patches

Human bone-marrow MSCs (passage 1-6) purchased from ATCC were first cultured in MesenPRO RS™ Medium until reaching 70% confluency in T-75 flasks. MesenPRO RS™ Medium has been specifically formulated by Gibco for growing human MSCs while maintaining their multipotential phenotype. Then, the cells were washed three times with PBS to remove excess serum, and the medium was replaced with EV-free media. To prepare EV-free MesenPRO RS™ Medium, the MesenPRO RS™ Growth Supplement was ultracentrifuged at 100,000 x *g* for 70 min at 4 °C and the supernatant was used to supplement the MesenPRO RS™ Basal Medium[108]. The conditioned medium containing MSC-EVs was collected after 48 h and the MSC-EVs concentrated via differential ultracentrifugation[109], resuspended in PBS, and quantified by nanoparticle tracking analysis (NTA)[40,44,110]. To fluorescently label the MSC-EVs, an NHS ester fluorophore (Dylight650, ThermoFisher Scientific) was added at a concentration of 0.1 mg/$10^{11}$ MSC-EVs, followed by incubation at 37 °C. The excess fluorophore was then removed using EV spin columns (molecular weight cutoff 3 kDa, ThermoFisher). MSC-EVs were then reconstituted in PBS at a concentration of $10^{11}$ EVs/ml. On each patch (air dried for 5 min), 100 μl of the fluorescent EV suspension was added to allow the EVs to conjugate for 30 min at 24 °C. The patches were then washed three times with PBS to remove unattached MSC-EVs. Patches were then cut into 4 sections and dipped in 24-well plates containing 250 μl of PBS, and the absorbance was measured at 647 nm using a plate reader. For confocal imaging (Fluoview, Olympus), the 647 nm laser was used to visualize the patch samples in PBS.

## EV release characterization

MSC-EVs were isolated as described above using differential centrifugation method[39,40,44] and labeled with DiD dye (Vybrant™ DiD from Thermofisher). For labeling, EVs were incubated in PBS with DiD at a 2.5 μM concentration for 1 h in dark at room temperature. The EVs were then washed three times with PBS by pelleting them via ultracentrifugation at 100,000 g and resuspending them in PBS. They were then loaded onto the patches as described above and then incubated at 37 °C in PBS. The control group was patches only with no EVs to measure background signal. The supernatant was collected at each time point and replenished with fresh PBS. The fluorescence of the

supernatant was measured using a plate reader. A standard curve of EVs vs fluorescence was generated to quantify the number of EVs that were released into the supernatant.

### In vivo lung puncture wound treatment with composite AuxES patches

12-week-old male SD rats were purchased from Charles River Laboratory. They were prepared as for the in vivo compliance studies (physiological ventilation volume of 7.5 ml/kg was used, IACUC Protocol No: 23-029.0). The tidal ventilation pressure in the airway was continuously monitored with a pressure transducer (Transpac IV, ICU Medical Inc). Pulmonary air leakage was induced in the right lower lobe of the lung by inserting an 18 G needle up to 1 cm deep in the lung tissue. Patches were placed over the injury site with forceps. Any changes in ventilation pressure were noted throughout the procedure.

### Declaration of Generative AI and AI-assisted technologies in the writing process

The authors used Microsoft Copilot to improve the readability of some parts of the manuscript. The authors reviewed and edited the content as needed and take full responsibility for the content of the publication.

### Reporting summary

Further information on research design is available in the Nature Portfolio Reporting Summary linked to this article.

## Data availability

Data from experiments conducted in this study are provided in the Article and Supplementary Information. Pathway enrichment analysis was performed using Panther and Reactome databases from pantherdb.org. The data underlying Fig. 1B, C; Fig. 2B; Fig. 3A; Fig. 4C–F; Fig. 5C,D; Fig. 6A, Fig. 7 A, B and Fig. 8B, E as well as Supplementary Figs. 1–3, 5, 6 are in the associated Source Data file. All data are available from the authors upon request. Source data are provided with this paper.

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

## Acknowledgements

J.N. received funding by the Eshelman Institute for Innovation at the UNC Eshelman School of Pharmacy, the National Science Foundation (NSF DMR 2000256) and the National Institutes of Health (R01GM150252, R01HL174038). Ameya Chaudhari is funded through the GlaxoSmith Kline Predoctoral Fellowship. Natalie Jasiewicz was funded through the PhRMA foundation predoctoral fellowship. Tom Egan and John Blackwell were supported by the UNC Lung Transplant Research Fund, with generous contributions by John Doherty and the Ferguson family, and the Cornelia D. Condon Memorial Fund for Lung Transplant Research. Some of the figures were created using Biorender.com and SolidWorks.

## Author contributions

P.C. and J.N. conceptualized the work. P.C., A.C., E.E., T.E., R.M., S.W., and J.N. designed the studies. P.C., A.C., E.E., E.B., M.H., J.L., M.M., C.K., J.B. R.S., and N.J. performed research. P.C., A.C., E.E., E.B., J.L., M.M., C.K., and J.B. analyzed the data. P.C. and J.N. wrote the original draft of the paper with edits and feedback from co-authors. P.C., A.C., J.N. edited and revised the paper. Resource and supervision: J.N.

## Competing interests

J.N., P.C., and T.E. are the inventors listed on the 2022 U.S. Provisional Patent Application No. 63/326,982, which is titled Adaptive Patches for Dynamic Organs. This patent application pertains to the patches that have been evaluated in the present study. These relationships have been disclosed to and are under management by UNC Chapel Hill. The remaining authors declare no competing interests.
