## [Peer Review File · Nature Communications]

Instantly adhesive and ultra-elastic patches for dynamic organ and wound repairReviewers' Comments:

Reviewer #1:

Remarks to the Author:

The manuscript entitled "Instantly adhesive and ultra-elastic patches for dynamic organ and wound repair" indeed exhibited many interesting biomedical applications of the presented adhesive patch. But it fails to show enough novelty to publish here. The most attractive part of this article is the frame structure of the super-adhesive hydrogel patch. However, it's already been published in August 2022 (Advanced Functional Materials, 32(43), 2207590). For another, the backbone polymer of the presented patch is GelMA/PAA, which had also been widely used to make adhesive substrates (Small 2022, 18, 2107544; New Journal of Chemistry, 38(7), 3112-3126). Besides, there are still some other comments are suggested to be address.

1. Some viewpoints are overdated or incorrect in the introduction section. For example, 1) Other bioadhesives become ineffective and completely lose their tissue adhesiveness in wet environments or when they encounter bodily fluids; 2) Many other bioadhesive materials require at least several minutes of tight tissue-patch interfacing for effective tissue adhesion; 3) Furthermore, to avoid discomfort and pain from repeated dressing changes, patches for skin wound healing should be biodegradable and disintegrate without causing a foreign body response; 4) Integration of therapeutics and drugs into liquids, glues, and hydrogels can, however, be challenging due to their inability to retain the drugs. These above-mentioned issues had been got addressed very well in recent years, the authors should update the research progresses.
2. The authors state that the presented patch is a promising alternative to surgical sutures for wounds/cuts management. Wound suturing always needs a high density to avoid partial tissue eversion. However, the gap between the two lines of the patch looks too big for complete sealing.
3. The control group (no patch) of Figure 6 looks abnormal. This is an acute full-thickness skin defect model without anti-contraction rings and any scaffolds. So the wounds usually contract to a very small area within 7 days in this 6 mm skin defect model. While the wound size increased a lot on day 3. In addition, the photograph of the wound on day 6 shows there are some substrates on the surface of the wound.

Reviewer #2:

Remarks to the Author:

The manuscript entitled "Instantly adhesive and ultra-elastic patches for dynamic organ and wound repair" by Chansoria et al, describes an elastic auxetic adhesive patch system for surgical uses. Especially, the authors aimed to achieve (i) instantaneous and strong tissue adhesion; (ii) ultra-elasticity, (iii) adaptable and scalable printing into precise anisotropic-auxetic architectures for a wide range of organs and tissues, (iv) biocompatibility and biodegradability, and (v) ability to deliver therapeutics. This is highly innovative research that has an urgent unmet need for surgical applications. The material has strong adhesion to soft tissues and is conformable to tissue movements. The auxetic designs provide additional flexibility and adaptability to different potential applications outlined in the manuscript including cardiac patch immobilizations, lung, and wound healing. It can be combined with therapeutics. The manuscript is an exciting advancement in the field.

One minor suggestion is to explore how the adhesion force changes in different GelMa-ACA ratios and also in different auxetic designs. The adhesion properties are closely related to the material's viscoelastic properties and the interface with the substrates.

It is a hydrogel-based system. The long-term swelling behavior and its impact on adhesion and mechanical behavior should be studied. Also, a study showing the impact of drying on adhesion and mechanical behavior of the adhesive patches should be added. This would be important for topical applications.

Reviewer #3:

Remarks to the Author:

In this manuscript, the authors introduce a highly elastic and adhesive patch for tissue repair. The patch exhibits strong and rapid adhesive performance, capable of adhering to organs nearly instantly. The authors designed patches with different shapes and structures due to the anisotropy and varying expansion/contraction ratios among organs. The in vivo experiment demonstrated good wound healing properties and the ability to repair lung puncture wounds, making this an interesting work. However, some key properties of the patch, especially under physiological conditions, are unclear. Therefore, the authors should address the following key issues.

1. In Figure 2C, the authors illustrated the major interactions between the patch and tissue. As ionic bonds (calcium bridges) are one of the most important interactions, do other metal cations in the buffer or blood, such as Na^+ and Mg^{2+} , affect their bonding? Additionally, if there is blood or other buffer solution on the tissue surface, will it affect the interaction between the patch and tissue? Furthermore, how does its adhesive performance hold up in dry conditions? Is water necessary for bonding?
2. As the patch for wound/tissue repair, what is the stability of the patch in buffer or blood? Will it dissolve or detach from tissue?
3. The authors loaded MSC-exosomes onto the patch to improve its wound healing ability. However, what is the stability of the exosome-loaded patch? Can it be stored at room temperature for an extended period? Additionally, how does the exosome release from the patch occur? It would also be useful to measure the drug release properties of the patch.
4. In Figure 6A, the wound size of the non-treated group showed a 1.5-fold increase on Day 1. Can you please explain this result? Additionally, MSC-exosomes played a major role in wound healing, as the patch without exosomes showed no treatment ability. What is the advantage of using the AuxES patch for wound healing? Its elastic and other properties do not seem to have a significant effect.
5. In the wound healing experiment, the patch was directly adhered to the wound. However, the other side of the patch that is exposed to the air remains sticky, which may cause it to adsorb dust or other particles from the air, leading to potential issues.
6. In the lung injury model, what about the long-term treatment effect of the patch? Will the patch degrade before the puncture wound is completely healed? Additionally, is there a risk of the patch adhering to other organs or blood vessels and causing further injury to the lung after application, given that the other side of the patch is still sticky?
7. The patch is shown to be highly elastic due to its specific auxetic architecture. However, such a structure may limit its applications for tissue repair. For example, its highly elastic properties may cause wound dehiscence, and the net structure may not prevent bleeding, as blood is easily able to flow out of the non-covered holes. Have the authors addressed these issues or proposed solutions for them? Additionally, what kind of biomedical application would be most suitable for the patch given these possible limitations?

Point-by-Point Response

We thank the reviewers for their constructive comments and suggestions. We have performed additional in vitro and in vivo experiments and significantly revised the manuscript based on the feedback. We hope that you agree that this has improved the manuscript. Please find our detailed responses below (references are provided at the end of this document). All major revisions are highlighted in **blue** in the revised manuscript and Supplementary Materials.

A brief overview of the additional, new experiments that have been performed have been listed below:

1. Effects of drying time on the adhesiveness (**Figure S1A**).
2. Effects of the change in concentration of acrylic acid on the bioadhesiveness of the patches (**Figure S1B**)
3. Effects of patch architecture on the bioadhesiveness (**Figure S1E**)
4. Release profile of MSC-derived extracellular vesicles (MSC-EVs) from patches (**Figure S3A**)
5. Biological activity of freshly isolated MSC-EVs compared to 1-week old MSC-EVs (**Figure S3B**)
6. Addition of a non-adhesive PEGDA layer over the patches demonstrate that the patches only demonstrate adhesion via the AuxES patch layer while the PEGDA layer demonstrates negligible adhesion (**Figure S5**).
7. New wound healing studies (**Figure 6A, B**) demonstrating superior wound healing with the AuxES patches coated with MSC EVs.
8. Swelling of patches after exposure to water, PBS or blood and control of swelling through post-curing (**Figure S6**).
9. Patches maintain their adhesiveness after exposure to blood (**Suppl. Video 10**).

Reviewer #1

Reviewer #1 (Remarks to the Author):

The manuscript entitled “Instantly adhesive and ultra-elastic patches for dynamic organ and wound repair” indeed exhibited many interesting biomedical applications of the presented adhesive patch. But it fails to show enough novelty to publish here. The most attractive part of this article is the frame structure of the super-adhesive hydrogel patch. However, it’s already been published in August 2022 (Advanced Functional Materials, 32(43), 2207590). For another, the backbone polymer of the presented patch is GelMA/PAA, which had also been widely used to make adhesive substrates (Small 2022, 18, 2107544; New Journal of Chemistry, 38(7), 3112-3126). Besides, there are still some other comments are suggested to be address.

We thank the reviewer for their comments. We would like to note that our study presents several innovations on a conceptual level and from a materials perspective, which are highlighted below:

- (1) Firstly, we would like to highlight that our current manuscript presents a completely new material that is ultra-stretchable and instantly adhesive at the same time. In contrast, our previous material¹ (Advanced Functional Materials, 32(43), 2207590) was not stretchable (yield strains <50%) or bioadhesive, and required fibrin glue for administration and was not bioadhesive. Fibrin glue is expensive (not suitable for low-income countries), is not effective at adhesion over wet tissues, and can carry the risk of infection. The material presented here addresses this limitation and is not only faster to fabricate but also features instant adhesion and ultra-stretchability, making it a landmark improvement over our previous work. As the new material in the submitted manuscript exhibits enhanced mechanics (higher stiffness and yield strains) compared to the previous material, we could also use yield strain of the different organs as an exclusion criterion (Figure 3D in the submitted paper). This would ensure that the patches do not break under the repetitive

motions of the dynamic organs. As a result, some of the meshes for the dynamic organs differ in their designs from the previous work. Also, we had not discussed any positive Poisson's ratio meshes in the previous work, which we demonstrate in the present work to cater to the mechanics of tissues such as diaphragm and tendons. Moreover, in the submitted manuscript, we have analyzed in detail the deformational characteristics of the different regions of the skin, and demonstrated that the patches could be tailored to different regions of the foot in dorsiflexion and plantarflexion, which has not been covered elsewhere. We have highlighted these facts in the introduction of the revised manuscript.

- (2) Secondly, the creation of a single material-based auxetic hole-filling patch is a novel concept, which is non-existent in literature. We demonstrate that the voids between the auxetic mesh elements can be filled with the same material as that used for the patch but at a lower concentration, which allows the patch to exhibit negative Poisson's ratio, while being an effective treatment of puncture wounds in dynamic organs (we demonstrate this using an in vivo Pneumothorax model). This is crucial for applications such as stopping bleeding and air leakage.
- (3) Thirdly, we would like to address the comparison with other articles on GelMA and synthetic polymer Polyacrylic acid (PAA) that the reviewer referred to. Our material formulation is based on a combination of GelMA and Acrylic Acid (ACA), post-treated with CaCl₂, which has a transformative impact on the properties of the patches. While the addition of CaCl₂ seems like a small change, it has transformative impact to the properties of the patches that has not been reported to date. We demonstrated in the manuscript how this treatment substantially improves bioadhesion, rendering unique bioadhesive and stretchable properties. Other distinctions with the studies, which reviewer referred to, are as follows:
 - Li et al.² (Small 2022, 18, 2107544) claimed bioadhesiveness, but their patches were not bioadhesive. They had to post-treat their constructs with NHS/EDC to mediate adhesion to tissues. Furthermore, all of their formulations had yield strains less than 20%, while our material features yield strains up to 100%, which is a 5-fold improvement compared to their material. Importantly, when combined with the unique auxetic architectures, our auxetic patches can feature yield strains up to 400%. Furthermore, the patches by Li et al. had elastic modulus between 0.4-1.2 kPa, while we demonstrate a broad range of elastic moduli (1 – 20 kPa), and stiffness ratios (1.0 to 4.0) and Poisson's ratios (-0.9 to 0.2), which allows our patches to cater to a broad portfolio of the dynamic organs.
 - Lastly, we would like to note that Serafim et al.³ (New Journal of Chemistry, 38(7), 3112-3126) did not show any tissue adhesiveness, only cell adhesion which is intrinsic to all GelMA-based matrices due to the presence of RGD peptides and not unique. Supporting this, we were able to show that our patch composition is substantially more adhesive than GelMA

We believe that this clarification helps to highlight the innovations presented in our manuscript.

Further comments from the reviewer: Some viewpoints are outdated or incorrect in the introduction section. For example:

- 1) Other bioadhesives become ineffective and completely lose their tissue adhesiveness in wet environments or when they encounter bodily fluids; 2) Many other bioadhesive materials require at least several minutes of tight tissue-patch interfacing for effective tissue adhesion;

We do understand the reviewer's concern here, and while we did not intend to make general comments, it may have been perceived to be so. We have altered these viewpoints in the introduction, and rather replaced them with advancements in the materials which demonstrate the instantaneous bioadhesion for prolonged periods in wet conditions. Specifically, we have added the following sentences in the introduction:

“While earlier bioadhesive materials required at least several minutes of tight tissue-patch interfacing for effective tissue adhesion^{8,9} (refs in updated manuscript file), recent advances have demonstrated quicker and more rapid bioadhesion to dry and wet tissues^{10,11} (refs in updated manuscript file). However, barriers to clinical translation of bioadhesive materials still exist and include insufficient flexibility and an inability to conform to the mechanics of internal organs and the skin.” – Page 3 of Main Text

In the introduction, we have also focused on the synergy of the new bioadhesive-elastic printable material with the framework for catering the patch designs to the dynamic organs, which will be difficult to implement with existing instantly adhesive or elastic materials as they are mostly not compatible with additive manufacturing.

3) Furthermore, to avoid discomfort and pain from repeated dressing changes, patches for skin wound healing should be biodegradable and disintegrate without causing a foreign body response;

We have changed the wording now to “Furthermore, to avoid discomfort and pain from repeated dressing changes, patches should also be made to biodegrade and disintegrate during the period of administration^{4,5}.”

4) Integration of therapeutics and drugs into liquids, glues, and hydrogels can, however, be challenging due to their inability to retain the drugs. These above-mentioned issues had been got addressed very well in recent years, the authors should update the research progresses.

We have removed the sentence stating the “inability [of hydrogels] to retain the drugs”, and updated the research progress by including seminal reviews which have highlighted and discussed hydrogels or hydrogel-based patches for drug delivery^{6,7}.

2. The authors state that the presented patch is a promising alternative to surgical sutures for wounds/cuts management. Wound suturing always needs a high density to avoid partial tissue eversion. However, the gap between the two lines of the patch looks too big for complete sealing.

We would like to note that the last part of the paper involves a demonstration of hole-filling patches which could be used for treatment of puncture wounds. Here, just a single material can be used to engineer the patches which can exhibit auxetic characteristics, while also effectively preventing fluid leakage. The auxetic meshes presented in this work can serve different purposes: firstly as a depot for therapeutic delivery (we demonstrate this using the extracellular vesicle-laden patches for cutaneous wound healing), or secondly, as a cell-adhesive substrate (GelMA contains RGD peptides) for cell proliferation. We do realize that the patches may not be able to replace surgical sutures or staples, and to address the reviewer comment, we have removed the sentence from the introduction.

3. The control group (no patch) of Figure 6 looks abnormal. This is an acute full-thickness skin defect model without anti-contraction rings and any scaffolds. So, the wounds usually contract to a very small area within 7 days in this 6 mm skin defect model. While the wound size increased a lot on day 3. In addition, the photograph of the wound on day 6 shows there are some substrates on the surface of the wound.

In terms of wound contraction within 7 days: Published studies report average wound closure in untreated controls in the range of 14.85% - 75%⁸⁻¹⁰. In our repeated new wound healing (**Figure 6A, B**), we observed an average wound closure of the untreated wound to be between $35.8 \pm 18.97\%$ by day 7. Wounds treated with the “Re-en itr 4 AuxES Patch with CNP and MSC-EVs” showed superior wound healing compared to untreated wound and wounds treated with ‘MSC-EVs Only.’

Response Figure 1. Comparing average % wound closure by day 7 reported in representative previously published studies⁸⁻¹⁰ to the extent of wound closure we observed in our study. Statistical analysis was performed using a student's t-test and the two groups were not significantly different.

In terms of what the reviewer refers to as “substrate” on the surface of the wounds: Based on the pathology report prepared by our co-author Dr. Rani Sellers – an experienced pathologist - the substrate on the wound surface is part of a serocellular crust (scab) - comprised of serum proteins and inflammatory cell infiltrates - which is expected in normal skin wound healing.¹¹

Wounds treated with the “Re-en itr 4 AuxES Patch with CNP and MSC-EVs” showed substantially improved wound contraction with minimal or no scab formation, suggesting that the AuxES hydrogel patches provides critical moistness to the wounds while the MSC-EVs promotes wound healing (**Main Text Figure 6 A, B**).

Notably, our control wounds remained untreated, as our patches are self-adherent and do not require additional supportive dressings for application. Unlike many standard wound dressings, such as the Integra® Bilayer Matrix Wound Dressing, or Tegaderm, our patches naturally degrade over time. Consequently, no wound dressing changes are necessary. Traditional dressing changes can cause pain and bleeding at the wound sites. Moreover, unlike the Integra® Bilayer Matrix Wound Dressing, our patches do not require stapling or suturing for secure attachment. This unique feature minimizes the risk of bleeding, skin tearing, and discomfort associated with dressing changes.^{12,13}

Main Text Figure 6 A, B: AuxES patches can be an effective treatment for wound healing. A. Re-en itr1 and Re-en itr4 AuxES patches laden with CNPs and MSC-EVs demonstrated significantly greater

wound closure compared with other treatment groups after a week of administration over cutaneous wounds in mice. *indicates $p < 0.05$ (group-wise comparisons by Tukey's HSD post-hoc tests). B. Photomicrographs of the margin of skin wound. Re-epithelialization (represented by the arrow \uparrow) and granulation tissue are indicated in the figure. Hematoxylin and eosin (H&E) staining, 20x objective (scale bar = 50 μm).

In our new Figure 6 A, B we performed the wound healing studies for 7 days where we see substantially improved wound contraction of $68.7 \pm 20.47\%$ for the group treated with "AuxES patches with CNP and MSC EVs". In comparison, for the non-treated control we see wound contraction of $35.8 \pm 18.97\%$. This is within the range of reported wound contractions for untreated wounds (see Response Figure 1).⁸⁻¹⁰

Additionally, we are including Tegaderm control studies (Response Figure 2) to demonstrate that our wound healing studies yield similar results to those published.¹⁴⁻¹⁶ We observed that articles showing wound healing studies without scab formation and enhanced wound contraction utilized Tegaderm. Similarly, to those published studies we observed an average wound closure of $63.36 \pm 27.46\%$ by day 7 in the "Tegaderm Alone" group compared to $35.8 \pm 18.97\%$ in the "No Treatment" group.

Response Figure 2. A) Use of Tegaderm dressing in our wound healing model leads to enhanced wound closure and no scab formation compared to No Treatment Control (scale bar 5mm). B) Quantification of wound closure reveals average wound closure of $63.36 \pm 27.46\%$ by day 7 in the Tegaderm Alone group compared to $35.8 \pm 18.97\%$ in the "No Treatment" group. Statistical significance was calculated using two-way ANOVA, where **indicates $p < 0.01$ (group-wise comparisons by Šídák's multiple-comparisons test).

Reviewer #2

The manuscript entitled “Instantly adhesive and ultra-elastic patches for dynamic organ and wound repair” by Chansoria et al, describes an elastic auxetic adhesive patch system for surgical uses. Especially, the authors aimed to achieve (i) instantaneous and strong tissue adhesion; (ii) ultra-elasticity, (iii) adaptable and scalable printing into precise anisotropic-auxetic architectures for a wide range of organs and tissues, (iv) biocompatibility and biodegradability, and (v) ability to deliver therapeutics. This is highly innovative research that has an urgent unmet need for surgical applications. The material has strong adhesion to soft tissues and is conformable to tissue movements. The auxetic designs provide additional flexibility and adaptability to different potential applications outlined in the manuscript including cardiac patch immobilizations, lung, and wound healing. It can be combined with therapeutics. The manuscript is an exciting advancement in the field.

One minor suggestion is to explore how the adhesion force changes in different GelMa-ACA ratios and also in different auxetic designs. The adhesion properties are closely related to the material’s viscoelastic properties and the interface with the substrates.

Thank you for the suggestion. We have performed these additional experiments on the changes in adhesion properties with different GelMA and ACA ratios, where we show that the adhesion forces increase upon increasing the ACA concentration likely due to increased ionic interactions via Ca^{2+} bridging and hydrogen bonding between the patch and the tissue (Added in the Supplementary Information as **Figure S1B**; entire Figure S1 is shown below).

As for the reviewer’s comment on exploring different designs, we evaluated the adhesion forces with 2 different designs. Here, the detachment forces are higher for patches which feature higher surface area of adhesion (now **Figure S1C, D, E** below). However, the detachment force normalized to the surface area was not significantly different which means that higher surface area of adhesion could allow greater linkages between the patch and the underlying tissue, thereby resulting in increased adhesion forces.

Supplementary Figure S1. A. Air drying of AuxES patches after CaCl_2 treatment improves the adhesiveness. B. The detachment force increased upon increasing the ACA content up to 20%, which can be attributed to increased ionic interactions via Ca^{2+} bridging and hydrogen bonding between the patch and the tissue. C. Comparison of the bioadhesiveness of the different patch architectures, where Re-en itr7 and Sin itr1 architectures (15×15 mm²) were

selected. **D.** Adhesiveness of the patches was compared in a tensile testing setup. **E.** Re-ent it7 patches with higher surface area of attachment demonstrated a higher attachment force compared to the Sin itr1 patches (* represents $p < 0.05$), but the force normalized to the surface area was not significantly different. Differences in the means of the normalized detachment force could be attributed to the higher elasticity of the Re-ent it7 patches compared to the Sin itr1 patches.

It is a hydrogel-based system. The long-term swelling behavior and its impact on adhesion and mechanical behavior should be studied. Also, a study showing the impact of drying on adhesion and mechanical behavior of the adhesive patches should be added. This would be important for topical applications.

We did observe changes to the adhesion properties of the patches upon air drying. We have performed additional data analysis with replicates (see new Supplementary **Figure S1 A, B**; shown above) to show that air drying can allow the patches to exhibit increased adhesiveness for wet tissue application, i.e. the application to the lungs. This may be due to increased water adsorption capacity and hydrogen bond formations between the tissue and the patch, however no air drying was necessary for application on dry tissue such as the skin.

Our hydrogel-based patches swell substantially when incubated in PBS (new results are demonstrated in **new Supplementary Figure S6 B**). Here, we have performed additional experiments where we show that the swelling of the patches can be controlled through post-curing the washed-up patches. This is done through incubation in PBS containing the photoinitiator followed by UV exposure to further crosslink the patch matrix. For instance, post-curing could limit the swelling of the patches to up to 50% in blood, compared to over 200% without post-curing.

Supplementary Figure S6 B. Analysis of the size of the patches after swelling relative to the original patch size (100% refers to the patch maintaining its original size upon incubation in the respective medic condition). Here, the post-cured patches demonstrated up to 50% and 80% increase in size upon incubation in blood and PBS, respectively, while the uncured patches swelled more than 200% in the respective conditions. A swelling of up to 80% is comparable to the hydrogel systems used for patches. The DI water patches did not demonstrate any swelling, which could be attributed to the absence of competing ions which could disrupt the calcium crosslinks in the patch network.

Reviewer #3 (Remarks to the Author):

In this manuscript, the authors introduce a highly elastic and adhesive patch for tissue repair. The patch exhibits strong and rapid adhesive performance, capable of adhering to organs nearly instantly. The authors designed patches with different shapes and structures due to the anisotropy and varying expansion/contraction ratios among organs. The in vivo experiment demonstrated good wound healing properties and the ability to repair lung puncture wounds, making this an interesting work. However, some key properties of the patch, especially under physiological conditions, are unclear. Therefore, the authors should address the following key issues.

1. In Figure 2C, the authors illustrated the major interactions between the patch and tissue. As ionic bonds (calcium bridges) are one of the most important interactions, do other metal cations in the buffer or blood, such as Na⁺ and Mg²⁺, affect their bonding? Additionally, if there is blood or other buffer solution on the tissue surface, will it affect the interaction between the patch and tissue? Furthermore, how does its adhesive performance hold up in dry conditions? Is water necessary for bonding?

The reviewer has raised a valid point. Extensive characterization of the changes to the physical properties of the patches such as the patch size and adhesion upon exposure to blood and its constitutive serum proteins and ions will be a scope of investigation of our future studies. In the experiments within the manuscript, excess blood over the organs was removed through gentle dabbling to dry-up the organ surface (we have highlighted this in the results section (it was earlier a part of the methods)). However, to answer the points raised by the reviewer, we have performed additional experiments to characterize the bioadhesiveness of the patches after exposure to blood, where we demonstrate that the patches still retain their adhesive characteristics (see **new Supplementary Video S10**; snapshots of the video are added below in **response Figure 3**) for a short-term exposure. However, swelling of the patches in blood during long-term incubation may affect this adhesion characteristics. For instance, our current AuxES hydrogel-based patches swell up to 200% in size when incubated in blood for 24 h at 37°C (see **new Supplementary Figure S6**, corresponding data was shown above for the comment by the previous reviewer). We have demonstrated that this swelling can be controlled through post-curing the washed-up patches. This is performed after washing the patches after fabrication and incubating them in water containing the photoinitiator, followed by UV exposure to further crosslink the patch matrix (we have added this in the methods). Here, we demonstrate that the post-curing could limit the swelling of the patches to up to 50% in blood (**new Figure S6**), which could be a useful strategy for future work on long-term survival studies with the AuxES patches.

We have added these considerations in the discussion under the following text – Page 18 of Main Text
*“While our AuxES patches have demonstrated adhesive characteristics when exposed to blood (**Supplementary Video S10**), long-term incubation in blood may potentially affect their adhesion properties due to patch swelling. For instance, our current AuxES hydrogel-based patches can swell up to 200% in size compared to the original (10×10 mm²) after 24h incubation in blood at 37°C (**Figure S6**). We have successfully demonstrated that this swelling can be controlled through post-curing the patches. Post-curing involves washing the fabricated patches, followed by incubation in water containing the photoinitiator, and subsequent exposure to UV light to enhance crosslinking of the patch matrix. Through this approach, we were able to reduce patch swelling in blood to a maximum of 50% increase in size compared to the original (**Figure S6**).”*

Response Figure 3. Snapshots from the video demonstrating that the organs still exhibit attachment over the patches exposed to blood or blood followed by PBS.

2. As the patch for wound/tissue repair, what is the stability of the patch in buffer or blood? Will it dissolve or detach from tissue?

We have demonstrated through additional experiments that short-term exposure of the patches to blood seems to retain the adhesiveness of the patches (**Supplementary Video S10**; snapshots of the video were shown in the **Response Figure 3**). However, long-term exposure to blood or buffer such as PBS can induce swelling of the patches, which can affect its physical properties including adhesion and that post-curing may be needed to prevent excessive swelling and loss of adhesiveness (**new Supplementary Figure S6 B**). As for the longevity of the current patch system, we expect that this patch should dissolve over a week or two *in vivo*, as was evidenced in the mouse models for wound healing. However, secondary crosslinking of the patches may further increase this degradation timeline of the patches.

3. The authors loaded MSC-exosomes onto the patch to improve its wound healing ability. However, what is the stability of the exosome-loaded patch? Can it be stored at room temperature for an extended period? Additionally, how does the exosome release from the patch occur? It would also be useful to measure the drug release properties of the patch.

We have performed additional experiments to characterize the release profiles of exosomes from the patches. These results have been added as a new **Supplementary Figure S3**. The exosomes demonstrated an initial burst release within the first 12 h, followed by a slow sustained release over a week in culture. Exosomes collected from patch after a week of incubation also demonstrated similar

effectiveness for in vitro wound closure in scratch assays as compared to freshly collected exosomes from MSCs.

Supplementary Figure S3. A. Release of MSC-EVs from the patches based on the estimation of the number of MSC-EVs being released, which demonstrates an initial burst release within the first 12 h, followed by a slow sustained release over a week in culture. **B.** Results from scratch assays with 3T3 cells, where the MSC-EVs collected from patch after a week of incubation demonstrated similar effectiveness for in vitro wound closure when compared to freshly collected MSC-EVs.

4. In Figure 6A, the wound size of the non-treated group showed a 1.5-fold increase on Day 1. Can you please explain this result? Additionally, MSC-exosomes played a major role in wound healing, as the patch without exosomes showed no treatment ability. What is the advantage of using the AuxES patch for wound healing? Its elastic and other properties do not seem to have a significant effect.

We apologize for this discrepancy; we mistakenly used a lower magnification image for the wound site of the control group on Day 0. This has now been revised. We have repeated the study and added another control 'MSC-EVs Only' and 'Re-en itr4 AuxES Patch with CNP and MSC-EVs' group (**Figure 6A, B**). We showed that our AuxES patches loaded with MSC-EVs performed significantly better than the 'MSC-EVs Only' group, underscoring the importance of a delivery device for achieving high therapeutic efficacy. The wound healing example primarily served to demonstrate that the patches can function as an effective depot and matrix for delivering therapeutic substances (such as exosomes). The advantages of using the AuxES patches for wound healing lie in their self-adhesion, ability to degrade, and ability to carry and release MSC-EVs. Unlike many standard wound dressings, such as the Integra® Bilayer Matrix Wound Dressing, or Tegaderm, our patches naturally degrade over time. Consequently, no wound dressing changes are necessary. Dressing changes can cause pain and bleeding at the wound sites. Moreover, unlike the Integra® Bilayer Matrix Wound Dressing, our patches do not require stapling or suturing for secure attachment. This unique feature minimizes the risk of bleeding, skin tearing, and discomfort associated with dressing changes.^{12,13,17}

In clinical scenarios, conformation to the skin mechanics and areas of high motion and flexing (i.e., knee and elbow) would improve mobility of the patient. In cases of more serious injuries such as myocardial infarction, such a conformation could also prevent undue stresses on the infarct,¹⁸ which could otherwise lead to pathological remodeling. We have highlighted these aspects in the discussion accompanying the wound healing results.

5. In the wound healing experiment, the patch was directly adhered to the wound. However, the other side of the patch that is exposed to the air remains sticky, which may cause it to adsorb dust or other particles from the air, leading to potential issues.

Thank you for raising this important point. We would like to address this concern together with the subsequent point raised by the reviewer of the patches sticking to other organs of the body. The accumulation of dust or other particles could be mitigated by coating the AuxES patches with a thin breathable layer of bandage such as Tegaderm®, but it might restrict the movement at the wound site. Here, a non-fouling coating over the patches, which could still allow maintenance of patch mechanics would be ideal. Towards this, we have performed additional experiments (see new **Supplementary Figure S5 B, C**), where we demonstrate how the top part of the patches could be coated with the layer of PEGDA which could potentially act as a non-adhesive layer. In the future, we plan to investigate additional non-fouling and non-adhesive materials and identify formulations which could allow the patches to retain their mechanics, while also demonstrating instantaneous and strong adhesion on one side.

Supplementary Figure S5. B. fabricated PaxS patches (FITC labeled) with a non-adhesive PEGDA layer (Rhodamine layer). **C.** The patches only demonstrate adhesion via the AuxES patch layer while the PEGDA layer demonstrates negligible adhesion.

We have also added these considerations in our discussion as the following text – Page 18 of Main Text “when using these patches for internal organ applications, it is important to consider the potential risk of the patch surface adhering to unintended organs or blood vessels, which could lead to further injury in the surrounding regions. To address this concern, we have conducted additional experiments (**Figure S5**), focusing on the development of a non-fouling coating for the patches while preserving their mechanical functionality. In these experiments, we coated the top part of the patches with polyethylene glycol diacrylate (PEGDA), which serves as a non-adhesive layer. To facilitate differentiation and imaging of the PEGDA and AuxES patch layers, we utilized rhodamine and fluorescein isothiocyanate (FITC) labeling, respectively. Our findings clearly demonstrate that the patches exhibit adhesion exclusively through the AuxES patch layer, while the PEGDA layer shows negligible adhesion.”

6. In the lung injury model, what about the long-term treatment effect of the patch? Will the patch degrade before the puncture wound is completely healed? Additionally, is there a risk of the patch adhering to other organs or blood vessels and causing further injury to the lung after application, given that the other side of the patch is still sticky?

The author has raised very valid points. Investigation of the long-term treatment effect of the patch system requires survival studies which would be a future scope of our investigation as that would require new ethical approvals, and financial and resource allocations. Furthermore, the property of

the patch to adhere to other organs may not be a major concern in the context of the lung puncture model, as the adhesion of the patch to the pleural surface of the chest can help with prevention of pneumothorax. However, stickiness of the patch to other organs may be a point of concern for other pathologies such as hemorrhage. Here, the adhesion of the patch to other organs could be mitigated through addition of a non-adhesive layer. We have performed additional experiments where we coated one side of the patch using PEGDA to prevent it from sticking to the other part (new **Supplementary Figure S5**).

7. The patch is shown to be highly elastic due to its specific auxetic architecture. However, such a structure may limit its applications for tissue repair. For example, its highly elastic properties may cause wound dehiscence, and the net structure may not prevent bleeding, as blood is easily able to flow out of the non-covered holes. Have the authors addressed these issues or proposed solutions for them? Additionally, what kind of biomedical application would be most suitable for the patch given these possible limitations?

In addition to demonstrating the use of auxetic meshes for conforming to the dynamics of the different organs, we also demonstrated how the patches could be used for the treatment of fluid leakage, using an in vivo rat model of lung puncture injury. Here, we presented a single material-based hole filling auxetic patches, where the holes were made of the same sticky and elastic material but the material is softer. This allowed the patch to maintain its auxetic mechanics, while preventing air leakage from the wound site.

We have also added the following points in the discussion – Page 19 of Main Text

“A hole-filling patch system can potentially also be used^{19,20}, such as those inflicted during hemorrhage. Notably, the materials may need fine-tuning for allowing patch adhesion over a continuously bleeding site. As for the auxetic patches with open holes, potential applications include the delivery of therapeutics for diabetic chronic wound healing^{21,22} or treatment of myocardial infarction^{18,23} or stomach ulcers^{24,25}. Furthermore, the crosslinking mechanics and material constitution of the patches could be fine-tuned to match the actual stiffness of the organs, where the patches could then be used to provide mechanical support to the organ. This is especially important to reduce wall stresses at the infarct and border zones in the heart, following an ischemia reperfusion injury^{23,26}.”

References

1. Chansoria, P. *et al.* Rationally Designed Anisotropic and Auxetic Hydrogel Patches for Adaptation to Dynamic Organs. *Adv. Funct. Mater.* 2207590 (2022) doi:10.1002/ADFM.202207590.
2. Li, W. *et al.* A Shape-Programmable Hierarchical Fibrous Membrane Composite System to Promote Wound Healing in Diabetic Patients. *Small* **18**, 2107544 (2022).
3. Serafim, A. *et al.* One-pot synthesis of superabsorbent hybrid hydrogels based on methacrylamide gelatin and polyacrylamide. Effortless control of hydrogel properties through composition design. *New J. Chem.* **38**, 3112–3126 (2014).
4. Dolan, E. B. *et al.* A bioresorbable biomaterial carrier and passive stabilization device to improve heart function post-myocardial infarction. *Mater. Sci. Eng. C* **103**, 109751 (2019).
5. Qu, M. *et al.* Biodegradable microneedle patch for transdermal gene delivery. *Nanoscale* **12**, 16724–16729 (2020).
6. Li, J. & Mooney, D. J. Designing hydrogels for controlled drug delivery. *Nature Reviews Materials* vol. 1 1–17 (2016).
7. Bernhard, S. & Tibbitt, M. W. Supramolecular engineering of hydrogels for drug delivery. *Adv. Drug Deliv. Rev.* **171**, 240–256 (2021).
8. Borges, P. A. *et al.* Adenosine Diphosphate Improves Wound Healing in Diabetic Mice Through

- P2Y12 Receptor Activation. *Front. Immunol.* **12**, 651740 (2021).
9. Kovner, A. V. *et al.* Wound healing approach based on excretory-secretory product and lysate of liver flukes. *Sci. Reports* **2022** *121* **12**, 1–15 (2022).
 10. Jiang, D. *et al.* MSCs rescue impaired wound healing in a murine LAD1 model by adaptive responses to low TGF- β 1 levels. *EMBO Rep.* **21**, (2020).
 11. Wilkinson, H. N. & Hardman, M. J. Wound healing: Cellular mechanisms and pathological outcomes. *Adv. Surg. Med. Spec.* 341–370 (2023) doi:10.1098/RSOB.200223/.
 12. Kirkcaldy, A. J., Wilson, M., Cooper, R., Baxter, S. K. & Campbell, F. Strategies for reducing pain at dressing change in chronic wounds: protocol for a mapping review. *BMJ Open* **13**, e072566 (2023).
 13. Gardner, S. E., Abbott, L. I., Fiala, C. A. & Rakel, B. A. Factors associated with high pain intensity during wound care procedures: A model. *Wound Repair Regen.* **25**, 558–563 (2017).
 14. Baldassarro, V. A. *et al.* Molecular mechanisms of skin wound healing in non-diabetic and diabetic mice in excision and pressure experimental wounds. *Cell Tissue Res.* **388**, 595–613 (2022).
 15. Savitri, C., Kwon, J. W., Drobyshava, V., Ha, S. S. & Park, K. M2 Macrophage-Derived Concentrated Conditioned Media Significantly Improves Skin Wound Healing. *Tissue Eng. Regen. Med.* **19**, 617–628 (2022).
 16. Thomas, H. M. *et al.* Plasma-polymerized pericyte patches improve healing of murine wounds through increased angiogenesis and reduced inflammation. *Regen. Biomater.* **8**, (2021).
 17. Fiala, C. A. *et al.* Severe pain during wound care procedures: A cross-sectional study protocol. *J. Adv. Nurs.* **74**, 1964–1974 (2018).
 18. Kapnisi, M. *et al.* Auxetic Cardiac Patches with Tunable Mechanical and Conductive Properties toward Treating Myocardial Infarction. *Adv. Funct. Mater.* **28**, 1800618 (2018).
 19. Yuk, H. *et al.* Rapid and coagulation-independent haemostatic sealing by a paste inspired by barnacle glue. *Nat. Biomed. Eng.* **2021** *510* **5**, 1131–1142 (2021).
 20. Haghniaz, R. *et al.* Tissue adhesive hemostatic microneedle arrays for rapid hemorrhage treatment. *Bioact. Mater.* **23**, 314–327 (2023).
 21. Augustine, R. *et al.* Growth factor loaded in situ photocrosslinkable poly(3-hydroxybutyrate-co-3-hydroxyvalerate)/gelatin methacryloyl hybrid patch for diabetic wound healing. *Mater. Sci. Eng. C* **118**, 111519 (2021).
 22. Pang, L. *et al.* In Situ Photo-Cross-Linking Hydrogel Accelerates Diabetic Wound Healing through Restored Hypoxia-Inducible Factor 1-Alpha Pathway and Regulated Inflammation. *ACS Appl. Mater. Interfaces* **13**, 29363–29379 (2021).
 23. Lin, X. *et al.* A viscoelastic adhesive epicardial patch for treating myocardial infarction. *Nat. Biomed. Eng.* **3**, 632–643 (2019).
 24. Ni, R. *et al.* Repairing gastric ulcer with hyaluronic acid/extracellular matrix composite through promoting M2-type polarization of macrophages. *Int. J. Biol. Macromol.* **245**, 125556 (2023).
 25. D'Argentre, A. D. P. *et al.* Programmable Medicine: Autonomous, Ingestible, Deployable Hydrogel Patch and Plug for Stomach Ulcer Therapy. *Proc. - IEEE Int. Conf. Robot. Autom.* 1511–1518 (2018) doi:10.1109/ICRA.2018.8460615.
 26. Li, Z. & Guan, J. Hydrogels for cardiac tissue engineering. *Polymers* vol. 3 740–761 (2011).

Reviewers' Comments:

Reviewer #1:

Remarks to the Author:

The authors have addressed my comments, I suggest this article can be accepted now.

Reviewer #2:

Remarks to the Author:

The new manuscript is thoroughly revised according to the reviewers' comments. Especially the new data on the different compositions of GelMA and ACA and the swelling behavior provide more thorough evidence of the short-term and long-term properties of the adhesive system. However, as reviewer #1 pointed out, the preceding work in another journal (*Advanced Functional Materials*, 32(43), 2207590) significantly impacts the novelty of this work. Although the authors claimed a few differences between the two technologies (including stretchability and difference in adhesive type), given that the key functional element is the auxetic structure rather than the stretchability itself, the preceding work already introduced the key concept. The application of the technology on foot is new and potentially has an implication for diseases such as diabetic foot ulcers, but the manuscript only introduces this to test different curvatures of the body.

Reviewer #3:

Remarks to the Author:

The authors have addressed all my critiques.

Point-by-Point Response

Reviewer #1 (Remarks to the Author):

The authors have addressed my comments, I suggest this article can be accepted now.

We would like to thank the reviewer for their time reviewing this manuscript and providing feedback.

Reviewer #2 (Remarks to the Author):

The new manuscript is thoroughly revised according to the reviewers' comments. Especially the new data on the different compositions of GelMA and ACA and the swelling behavior provide more thorough evidence of the short-term and long-term properties of the adhesive system. However, as reviewer #1 pointed out, the preceding work in another journal (Advanced Functional Materials, 32(43), 2207590) significantly impacts the novelty of this work. Although the authors claimed a few differences between the two technologies (including stretchability and difference in adhesive type), given that the key functional element is the auxetic structure rather than the stretchability itself, the preceding work already introduced the key concept.

Our patch system overcomes the limitations of the previously used material. The material reported earlier requires the use of fibrin glue to attach the patch to tissues. However, fibrin glue is expensive and hinders the application of our technology in low-resource settings. Our patch material exhibits instant adhesion, high elasticity, and can be printed at a rapid rate, all of which enhance the clinical translatability of this system. Furthermore, this paves the way for applications, such as wound healing, as we have shown in Figure 6.

Additionally, we developed novel hole-filling auxetic patches using the same materials as the lattice structure to fill the voids, all while retaining the auxetic nature of the patches. Using the same material to fill the voids presents a significantly simpler and more cost-effective alternative to commonly used sealants such as tissue glues (like Fibrin). These void-filled AuxES patches retained their auxetic nature and were able to restrict pulmonary air leakage (Figure 8).

To address the reviewers' comment, the following paragraph has been added to the introduction:

“Furthermore, the use of fibrin glue for adhesion could introduce variability during the preparation and application of the glue, potentially affecting the mechanics of the patches. For consistent clinical implementation, prefabricated adhesive patches are more suitable, as they can be prepared in a controlled environment and provided to the clinics for immediate use.”

The application of the technology on foot is new and potentially has an implication for diseases such as diabetic foot ulcers, but the manuscript only introduces this to test different curvatures of the body.

As shown in Figure 6, our studies extend beyond testing different curvatures of the body. The AuxES patches coated with extracellular vesicles derived from mesenchymal stem cells demonstrate robust wound healing capability in vivo without inducing a foreign body response. Furthermore, our patches are designed to degrade naturally, eliminating the need for dressing changes which could cause bleeding, skin tearing, and other related injuries.

This has been addressed in the discussion section as follows:

“Notably, AuxES patches demonstrated gradual dissolution into the skin after a week (Figure 6A) and do not need to be removed. A degradable patch is desirable as it eliminates the need for dressing changes, which could lead to bleeding, skin tearing, and other related injuries.”

Additionally, we have elaborated on the discussion regarding potential future applications of the AuxES patches:

“A void-filling patch system can potentially also be used for the treatment of bleeding wounds,^{93,94} such as those inflicted during hemorrhage. Notably, the materials may need fine-tuning for allowing patch adhesion over a continuously bleeding site. As for the auxetic patches with open holes, potential applications include the delivery of therapeutics for diabetic chronic wound healing^{95,96}, burn wounds, treatment of myocardial infarction^{12,68} or stomach ulcers^{97,98}. Our future studies will investigate the use of these patches laden with or without therapeutic substances (e.g., small molecules, larger proteins or EVs) for treating these pathologies. Additionally, we plan to investigate whether the crosslinking mechanics and material constitution of the patches can be fine-tuned to match the actual stiffness of the organs, where the patches can be used to provide mechanical support to the organ (e.g. cardiac patches). This is especially important to reduce wall stresses at the infarct and border zones in the heart, following an ischemia reperfusion injury^{68,99}.”

Reviewer #3 (Remarks to the Author):

The authors have addressed all my critiques.

Thank you for your time reviewing the manuscript and providing feedback.